# The type III effector NopL interacts with GmREM1a and GmNFR5 to promote symbiosis in soybean

Chao Ma [1,2,9], Jinhui Wang [1,2,9], Yongkang Gao[3,9], Xulun Dong [2], Haojie Feng[2], Mingliang Yang[1,2], Yanyu Yu[2], Chunyan Liu [1] ✉, Xiaoxia Wu[2], Zhaoming Qi[2], Luis A. J. Mur [4], Kévin Magne [5,6], Jianan Zou[1], Zhenbang Hu[1], Zhixi Tian [7,8] ✉, Chao Su[3] ✉, Pascal Ratet [5,6] ✉, Qingshan Chen [1,2] ✉ & Dawei Xin [1,2,4,5] ✉

The establishment of symbiotic interactions between leguminous plants and rhizobia requires complex cellular programming activated by Rhizobium Nod factors (NFs) as well as type III effector (T3E)-mediated symbiotic signaling. However, the mechanisms by which different signals jointly affect symbiosis are still unclear. Here we describe the mechanisms mediating the cross-talk between the broad host range rhizobia *Sinorhizobium fredii* HH103 T3E Nodulation Outer Protein L (NopL) effector and NF signaling in soybean. NopL physically interacts with the *Glycine max* Remorin 1a (GmREM1a) and the NFs receptor NFR5 (GmNFR5) and promotes GmNFR5 recruitment by GmREM1a. Furthermore, NopL and NF influence the expression of *GmRINRK1*, a receptor-like kinase (LRR-RLK) ortholog of the *Lotus* RINRK1, that mediates NF signaling. Taken together, our work indicates that *S. fredii* NopL can interact with the NF signaling cascade components to promote the symbiotic interaction in soybean.

Soybean (*Glycine max* (L.) Merr.), is an important legume crop and an important source of protein and cooking oil for humans and animals[1]. Chemical fertilizers are used extensively to achieve high soybean yields, paradoxically overlooking its symbiotic nitrogen-fixing capability[2,3]. An environmentally ecofriendly agronomic strategy should seek to better exploit rhizobial symbiosis with soybean for sustainable soybean production.

Symbiosis between legumes and rhizobacteria is established via mutual recognition and molecular interactions. In these processes,

flavonoids secreted by leguminous host plants induce the production of Nod factors (NFs) by rhizobia[4], which play a pivotal role in establishing the symbiotic interaction with legumes[5,6]. Host-specificity is determined by the chemical structure of the lipochito-oligosaccharide NFs[7–9]. *NodABC* genes, present in most *Rhizobium* species, are required for the synthesis of the core structure of NFs[10,11]. Other *Nod* genes introduce specific modifications on the NFs core structure to influence the relative symbiotic compatibility between legume hosts and different strains of rhizobia[4,10].

[1]College of Agriculture, National Key Laboratory of Smart Farm Technologies and Systems, Northeast Agricultural University, Harbin, China. [2]College of Agriculture, Key Laboratory of Soybean Biology in Chinese Ministry of Education, Northeast Agricultural University, Harbin, China. [3]Hubei Hongshan Laboratory, National Key Laboratory of Crop Genetic Improvement, College of Plant Science and Technology, Huazhong Agricultural University, Wuhan, China. [4]Department of Life Sciences, Aberystwyth University, Edward Llwyd Building, Aberystwyth, UK. [5]Université Paris-Saclay, CNRS, INRAE, Univ Evry, Institute of Plant Sciences Paris-Saclay (IPS2), Gif sur Yvette, France. [6]Université de Paris, Institute of Plant Sciences Paris-Saclay (IPS2), Gif sur Yvette, France. [7]Key Laboratory of Seed Innovation, Institute of Genetics and Developmental Biology, Chinese Academy of Sciences, Beijing, China. [8]University of Chinese Academy of Sciences, Beijing, China. [9]These authors contributed equally: Chao Ma, Jinhui Wang, Yongkang Gao. ✉e-mail: cyliucn@126.com; zxtian@genetics.ac.cn; chaosu@mail.hzau.edu.cn; pascal.ratet@universite-paris-saclay.fr; qshchen@126.com; dwxin@neau.edu.cn

Rhizobium NFs are perceived by symbiotic receptors that initiate a signaling cascade to cause root-hair infection and nodule organogenesis[12–14]. In lotus and soybean, the NF receptor complex is composed of NFR1 and NFR5, orthologs of *Medicago* LYK3 (lysin motif receptor-like kinase 3) and NFP (Nod Factor Perception)[15,16]. NFR1 and NFR5 form heterocomplexes where NFR5 interacts with SYMRK (SYMBIOSIS RECEPTOR-LIKE KINASE), linking NFs signaling to the common symbiosis signaling pathway (CSSP) used for root nodule symbiosis (RNS)[17,18]. Recognition of the NFs by the co-receptors triggers calcium spiking in the nucleus and phosphorylation of CYCLOPS/IPD3 by CCaMK/DMI3. This plays an important role in RNS specific gene expression, such as the key symbiotic gene *NIN*. The components of this signaling pathway (CSSP) are also used by arbuscular mycorrhiza to establish symbiotic interactions[19–21]. Remorins are plant-specific scaffolding proteins found in all land plant, including ferns and bryophytes, and are important regulators of plant-microbe interaction[22]. The *Medicago truncatula* symbiotic Remorin 1 (MtSYMREM1) is root nodule specific[23] and required for both rhizobia infection and bacterial release[24]. MtSYMREM1 interacts with NFP, LYK3 and MtSYMRK and required for the stability of entry receptor LYK3[23,25,26], but how these complexes contribute to signal transduction and specificity is not well understood.

Beside NFs, other rhizobial signaling molecules can influence symbiosis with legumes. The type III effectors (T3Es) secreted via type III secretion systems (T3SS) were previously described in plant pathogens[27–29] and now characterized in rhizobia[30,31]. In rhizobia, these T3Es, designated as Nodulation Outer Proteins (Nops), are delivered into the host cells to affect both infection and nodule formation. More than ten T3Es have been identified in rhizobia, including NopAA, NopC, NopD, NopI, NopL, NopM, NopP, NopT, InnB, ErnA and Sup3[31–34]. These T3Es can also play positive or negative roles in the establishment of symbiosis depending on the host species[31], especially, in the broad host range strains, including *Sinorhizobium fredii* HH103[35,36] and *Sinorhizobium sp*. strain NGR234[37,38]. NopL is a rhizobial-specific T3E, which is a substrate for plant kinases in vitro[39]. The *Sinorhizobium sp.* strain NGR234 NopL effector can be phosphorylated in *Nicotiana tabacum* by the Mitogen-activated protein kinase (MAPK) Salicylic Acid-Induced Protein Kinase (NtSIPK). NopL can also directly interacts with NtSIPK in onion and tobacco nuclei[40,41]. The *nopL* mutant of rhizobia or mutations in phosphorylation sites of NopL negatively affect the formation of *Phaseolus* nodules and promote nodule senescence[40]. However, it is not clear what the targets of NopL are in legumes or what the molecular mechanisms behind its function.

Previous studies have shown that T3Es can activate host nodulation signaling by bypassing NF recognition in some legume species, triggering similar events as NF-mediated symbiosis[32,42]. This suggests a potential overlap in T3Es and NF-mediated symbiotic signaling but the molecular mechanisms underlying the T3Es actions have yet to be described. We here show that NopL plays an essential role for the broad host range rhizobia HH103 in the establishment of NF-mediated symbiosis by interacting with GmREM1a and GmNFR5 to promote NF signaling in soybean.

## Results

### NopL is required for NF signaling in soybean
Previous studies have shown that some T3Es can hijack soybean NF signaling to promote rhizobium infection[34,43] and that NopL interacts with components of the plant immunity in non-legume plants[39,41]. Various NopL effector genes are encoded in a number of *Sinorhizobium* and *Bradyrhizobium* strains (Supplementary Fig. 1a, b). To investigate how the T3E NopL could affect NF-mediated symbiotic signaling in soybean, we constructed *nopL*, *nodA* (unable to synthesize NFs) mutants and the *nodAΩnopL* double mutant (Supplementary Fig. 2) in the *S. fredii* HH103 strain. Inoculation of the Suinong 14 (SN14) cultivar with the wild-type (WT) HH103 strain and the mutant strains

showed that fewer nodules were observed compared to the WT control. Whilst significantly different from controls, there were no significant differences in nodule number and dry weights in the *nopL*, *nodA*, and *nodAΩnopL* mutants at 28 days post-inoculation (dpi) (Fig. 1a, d and e). The same nodulation phenotype (reduced nodule number) was observed when SN14 was inoculated with *nodB*, *nodC* mutants and the double mutants with *nopL* (Supplementary Fig. 3a–e). To know if the rhizobium mutant phenotype could be plant genotype dependent, we tested a panel of soybean cultivars representative of the soybean pan-genome and significantly contributing to breeding and production[44]. The different genotypes developed a different number of nodules in the presence of the HH103 WT strain. In addition, they showed a similar reduced nodule number phenotype than SN14 when inoculated with *nopL*, *nodA*, and *nodAΩnopL* (Supplementary Fig. 3f). Hence, the absence of either NFs or NopL effector or both, did not block nodulation but only reduced its nodule number, suggesting that nodulation in soybean might be not strictly NF dependent, which is in agreement with previous work[42]. Furthermore, as there is no additive effect of the two mutations, it is likely that they are acting in the same pathway and that NF signaling requires NopL. Over-expression of the NopL protein in rhizobia (*NopL-GFP* fusion under the control of *nptII* promoter in HH103; Supplementary Fig. 2c) increased nodule numbers in SN14 (Supplementary Fig. 4a–c), indicating a positive effect on nodulation in SN14. A *NodApro:NodA* construct was used to restored a WT nodulation phenotype with *nodA* mutant but was inefficient in the *nodAΩnopL* double mutant (Supplementary Fig. 4d–f). This further suggested that NopL is strictly required for the NF-dependent symbiosis in soybean.

As NodQ is required to produce sulfated NF[45], to further understand NopL function, we generated *S. fredii* HH103ΩnodQ and *nodQΩnopL* mutants (Supplementary Fig. 2e). Interestingly, the *nodQ* mutants exhibited increased nodule numbers and weights compared to the WT strain indicating that the NF modification by NodQ reduces NF mediated activation of symbiosis (Supplementary Fig. 4g–i). In contrast there was no significant difference in the nodule number or dry weight between SN14 inoculated with *nopL* (reduced number of nodules) and *nodQΩnopL* mutants 28 dpi, suggesting again that NopL acts in the NF signaling cascade (Supplementary Fig. 4g–i). We next inoculated SN14 with WT HH103, *nopL*, *nodA,* and *nodAΩnopL* GUS-tagged strains. The number of infection threads (ITs) (1 dpi) and primordia (7 dpi) were similarly significantly reduced in the different mutants as compared to the WT strain (Fig. 1b, c, f, g), suggesting a positive role of NopL in promoting rhizobium infection, by facilitating the formation of ITs and the development of nodule primordia via the NF signaling cascade. Taken together, our results show that NopL plays an essential role in HH103 soybean symbiosis in mediating the NFs-dependent symbiotic signaling.

### NopL is delivered directly into soybean cells
In order to know if NopL is delivered in soybean plant cells, we studied its localization in root hair cells 1 dpi using WT or the corresponding *nopL* mutant. NopL protein was detected in the protein fraction of the nucleus and cell membranes (Fig. 2a). To further show that NopL is translocated to the soybean cells, a T3E-adenylate cyclase (Cya) reporter translocation assay was used. This showed that cAMP was detected in soybean roots inoculated with HH103 expressing a NopL-Cya fusion protein driven from the *NopL* promoter (HH103 NopL-Cya). No cAMP was detected in soybean roots inoculated with wild-type HH103 or with the *ttsI* mutant strain that is unable to translocate NopL-Cya protein to the plant cell (Fig. 2b).

As NopL in strain NGR234 is involved in later stages of nodulation[41], we also tested its localization in infected cells of functional nodules using immunohistochemistry (IHC) and immunofluorescence (IF) analyses. These studies indicated that NopL was delivered and was predominantly present in the nitrogen-fixing cells of

 

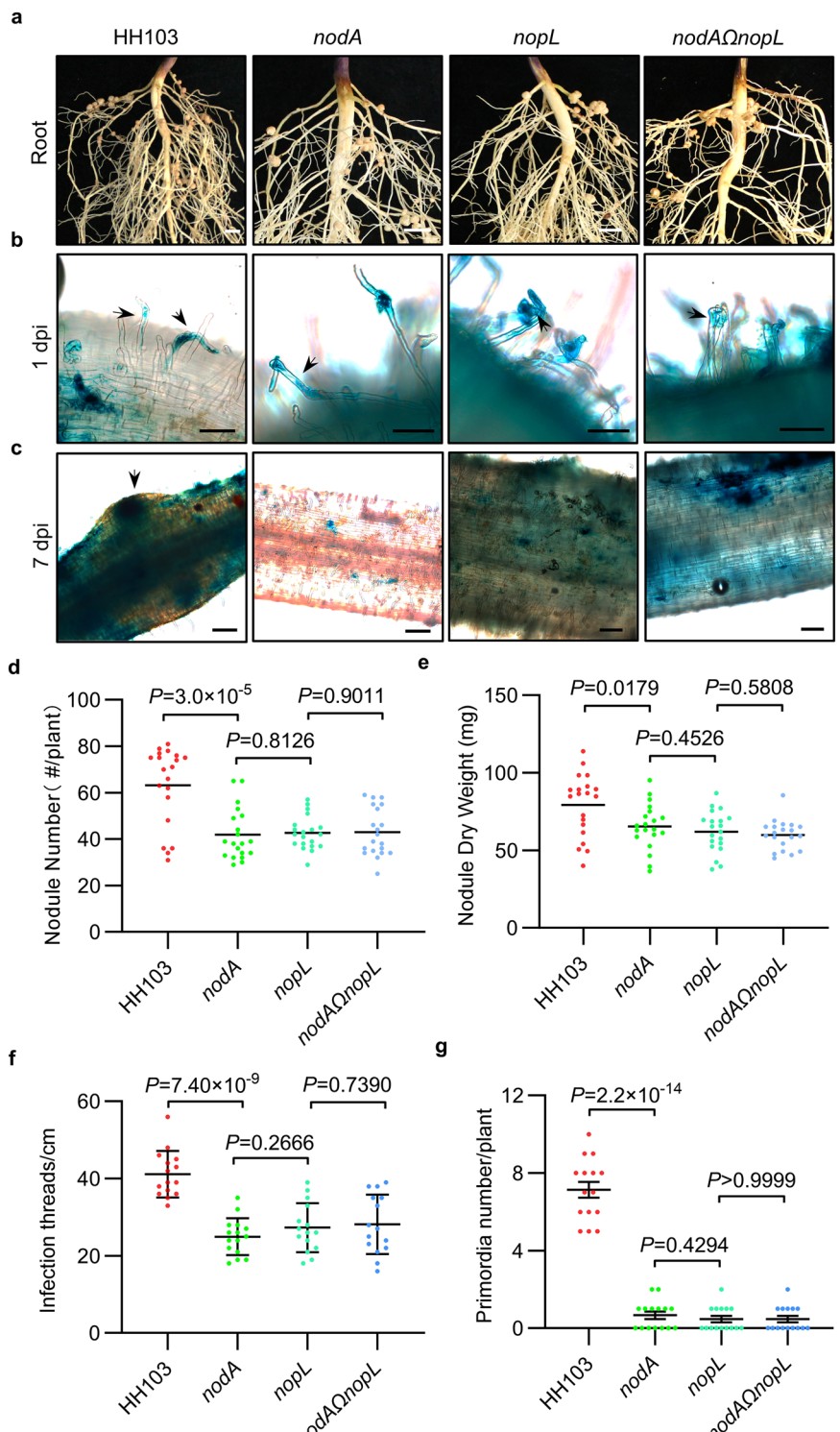

**Fig. 1 | The type III effector NopL promotes the NF signaling pathway in soybean. a** Roots of SN14 28 days post inoculation (dpi) with *Sinorhizobium fredii* HH103 (WT), *nodA* mutant, *nopL* mutant and *nodAΩnopL* double mutant. Scale bars = 5 mm. Infection thread and nodule primordia phenotypes of SN14 inoculated with GUS-tagged HH103 and its mutants 1 dpi (**b**) or 7 dpi (**c**). Scale bars = 100 μm in (**b**) and 200 μm in (**c**). **d** Nodule number per plant for (**a**) at 28 dpi (n = 20). **e** Nodule dry weight per plant for (a) at 28 dpi (*n* = 20). **f** Infection threads number per cm of root of SN14 for (**b**) at 1 dpi (Data are represented as mean ± SD, *n* = 15). **g** Nodule primordia number per SN14 plant for (**c**) at 7 dpi (Data are represented as mean ± SD, n = 15). Statistical analysis used Student's *t*-test (two-sided). Source data are provided as a Source Data file.

WT nodules (Fig. 2c–e, Supplementary Fig. 5a). Immunostaining with cyanine 3-conjugated IgG (Cy3-IgG) localized NopL to the cell membrane and the nucleus (Fig. 2f–h). This result was further refined using immuno-gold labeling by transmission electron microscopy and showed that NopL was associated to the membrane of the symbiosome as well as the nucleus and cytoplasm (Fig. 2i–k, Supplementary Fig. 5b). These results all suggest that NopL can be delivered into soybean root cells.

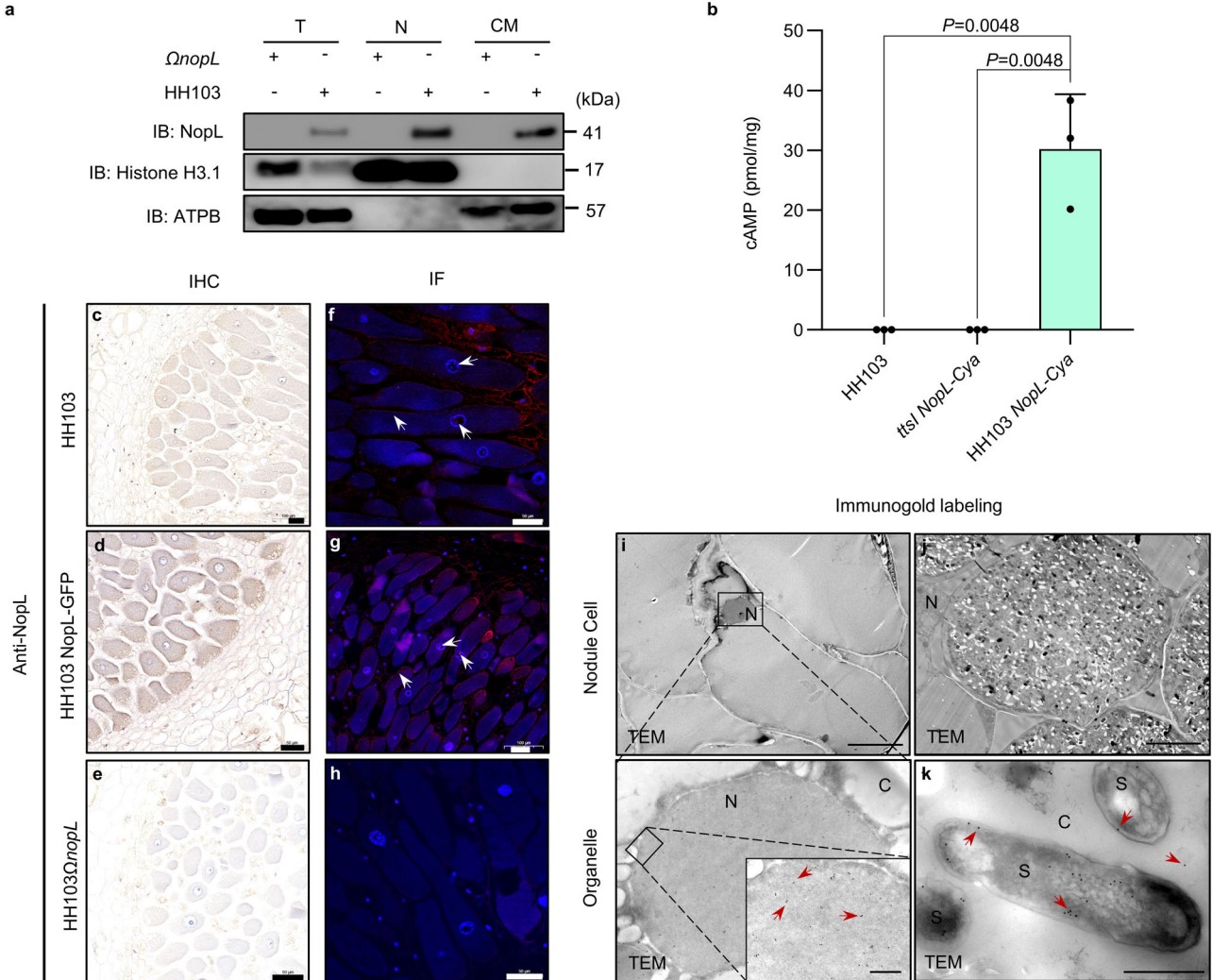

**Fig. 2 | The type III effector NopL is secreted into the host cell.**
**a** Immunolocalisation of NopL in the nucleus and cell membrane of root hair cells. Inoculation of *nopL* mutant as control. T, total protein. N, proteins of the cell nucleus. CM, proteins of the cell membrane. The ATP synthase protein (ATPB) was used as a control membrane protein in this experiment. **b** T3E- adenylate cyclase (Cya) reporter translocation assays shows that HH103 translocate NopL-Cya into soybean roots cells. Translocation was assayed based on cAMP production by the Cya reporter in soybean roots at 1 dpi inoculated with HH103, *ttsI NopL-Cya* and HH103 *NopL-Cya*. HH103 and *ttsI NopL-Cya* are included as a negative control. Data are represented as mean ± SD, and Statistical analysis used Student's *t*-test (two-sided, *n* = 3). **c**–**e** Immunohistochemistry (IHC) analysis of SN14 soybean root nodules 28 days post inoculation (dpi) with HH103, HH103(NopL-GFP) or

HH103Ω*nopL* mutant using Anti-NopL polyclonal antibodies. The nuclei were stained using hematoxylin. The *nopL* mutant was used as a control. **f**–**h** Immuno-fluorescence (IF) analysis of nodules 28 days post inoculation (dpi) with HH103, HH103 NopL-GFP or HH103Ω*nopL* using Anti-NopL polyclonal antibodies. IF images of nodule cells show NopL in the nucleus and cell membrane (red fluorescence) and DAPI in the nucleus (blue fluorescence). The *nopL* mutant was used as a control. White arrow, Cy3 signal in nodule cells. **i**–**k** Immunogold labeling of NopL in soybean nodule cells. N, nucleus. S, symbiosome. C, cytoplasm. Scale bars, 1 µm (nodule cell) or 500 nm (Organelle). Red arrows point to gold particles in nodule cell. The experiments in **i**–**k** were repeated three times with similar results. Source data are provided as a Source Data file.

## NopL physically interacts with GmREM1a

In order to identify the NF signaling cascade components targeted by NopL, a semi-pull-down assay was performed on protein extracts from isolated soybean root hairs (1 dpi) using a recombinant His-NopL protein. NopL-interacting proteins were identified by Liquid chromatography-tandem mass spectrometry (LC–MS/MS) (Supplementary Fig. 6a, b). A total of 246 candidate proteins was identified (Supplementary Data 2).

The candidates included a protein homologous to the remorin MtSymREM1[24,46], which was designed *Glycine max* SymREM1a (GmREM1a, Supplementary Fig. 6c). The interaction of NopL with GmREM1a was confirmed using Yeast two-hybrid (Y2H) (Supplementary Fig. 6d) and bimolecular fluorescence complementary (BiFC) analysis in *Nicotiana benthamiana* epidermal cells (Fig. 3a). Pull-down

analysis in vitro also indicated that NopL can directly interact with GmREM1a (Fig. 3b) as did in vivo co-immunoprecipitation (Co-IP) assays using soybean hairy roots (Fig. 3c). As soybean contains two SymREM1 protein homologs, GmREM1a and GmREM1b, with a high degree of similarity[46], we investigated whether NopL can also interacts with GmREM1b. BiFC analysis, in vivo Co-IP analysis and in vitro GST pull-down assays revealed that NopL can indeed interact with GmREM1b (Fig. 3a–c). All these results suggest that GmREM1a/b are targets of the NopL rhizobium effector during early steps of symbiosis.

NopL-GFP with GmREM1a and GmREM1b-RFP constructs were used to localize the protein complexes to the cell membrane in *N. benthamiana* cells (Supplementary Fig. 7a). NopL-GFP and GmREM1a/1b-RFP were no longer associated with the cell wall, and GmREM1a/1b did not alter the subcellular localization of NopL by plasmolysis with

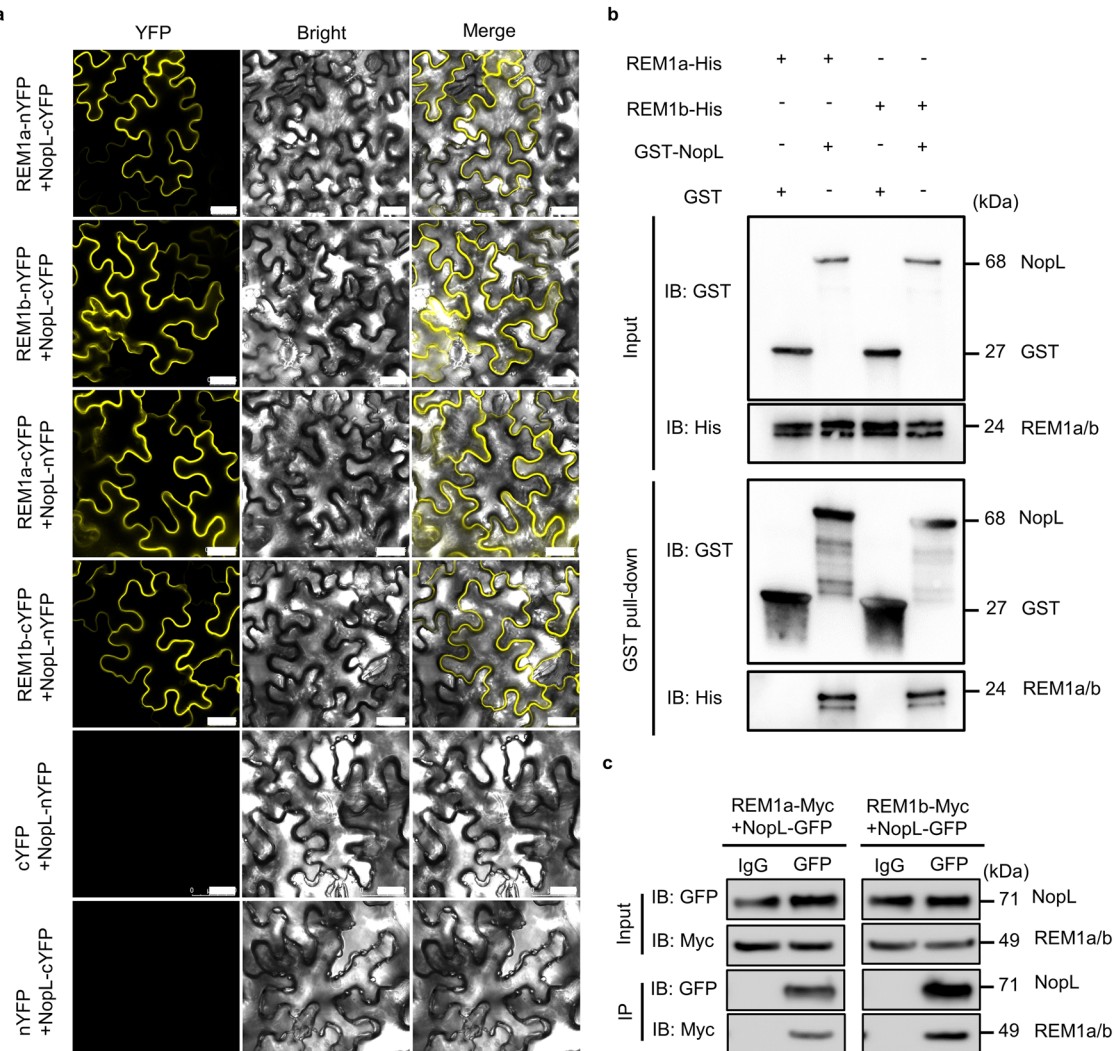

**Fig. 3 | NopL physically interacts with REM1a and REM1b. a** BiFC analysis of the interactions between NopL and GmREM1a (REM1a) and GmREM1b (REM1b). In the top and down panels, the split YFP is inversely fused in C or N position for the two proteins. Scale bars = 25 μm. **b** Interaction of NopL with REM1a and REM1b in in vitro GST pull-down assays. GST-NopL or GST proteins were used as baits. The antibody used for the pull-down assay are indicated on the left part of the panels. The size of the proteins in kilo Daltons (kDa) is indicated on the right side of the panel. IB: GST = Imunoblot using GST AB; IB: His = Imunoblot using His AB. **c** Co-IP assays showing NopL interaction with REM1a and REM1b in soybean hairy roots. The immunoprecipitation (IP) using the Anti-GFP antibody detects the interaction between NopL and GmREM1a and GmREM1b. The IgG antibody was used as a negative IP control. IB: Myc = Imunoblot using Myc AB; IB: GFP = Imunoblot using GFP AB. The experiments were repeated three times with similar results. Source data are provided as a Source Data file.

30% sucrose (Supplementary Fig. 7b, c). Immunofluorescence (IF) analysis using polyclonal antibodies to NopL and GmREM1a also showed that NopL (antibodies labeled with Cy3) co-localized with GmREM1a (antibodies labeled with FITC) on the plasma membrane in young root nodules (10 dpi) (Supplementary Fig. 7d–f). These data strongly suggest that NopL physically interacts with GmREM1a/b proteins in vivo.

## NopL-mediated NF signaling in soybean depends on GmREM1a

We noted that our GmREM1b was not targeted in our NopL protein semi-pull-down experiment (Supplementary Data 2), therefore we focused our analysis on GmREM1a. To investigate the role of GmREM1a/NopL interaction during NF signaling, we generated *Gmrem1a* knockout lines in the DN50 soybean genotype by CRISPR/Cas9. The three independent *Gmrem1a* knockout lines (*Gmrem1a-1* to *Gmrem1a-3*; Supplementary Fig. 8a, b) exhibited reduced (50%) nodule number and dry weights compared to the wild-type DN50 (Fig. 4a, b. Supplementary Fig. 8c) when inoculated with the WT HH103 rhizobium line. The knock-out phenotype could be

complemented following hairy root transformation with a construct expressing *GmREM1a* (*pGmREM1a:GmREM1a-GFP*; Supplementary Fig. 9a–c). This confirmed that the nodule phenotype of *Gmrem1a* was due to loss of GmREM1a function. Interestingly, contrary to what was observed after inoculation of the wild-type DN50, there was no significant difference in nodule number or dry weight in the *Gmrem1a* KO lines inoculated with the HH103 WT strain or the *nodA*, *nopL*, and *nodAΩnopL* mutants (Fig. 4a, b). We observed a similar phenotype in DN50 after gene silencing by RNAi of *GmREM1a* (Supplementary Fig. 8d, e). The analysis of the number of ITs in the *Gmrem1a* KO lines showed that there was also no significant difference in the number of ITs following inoculation with HH103 or the *nodA*, *nopL* or *nodAΩnopL* mutants (Fig. 4c, d). To further verify the GmREM1a-NopL interaction, GFP-tagged HH103 and HH103 strain overexpressing NopL-GFP were used to inoculate DN50 or *Gmrem1a* KO lines. Overexpression of *NopL* increased the nodule number in WT plants but did not change the number of root nodules in *Gmrem1a* KO lines (Fig. 4e, f) suggesting that GmREM1a is required for the NopL action.

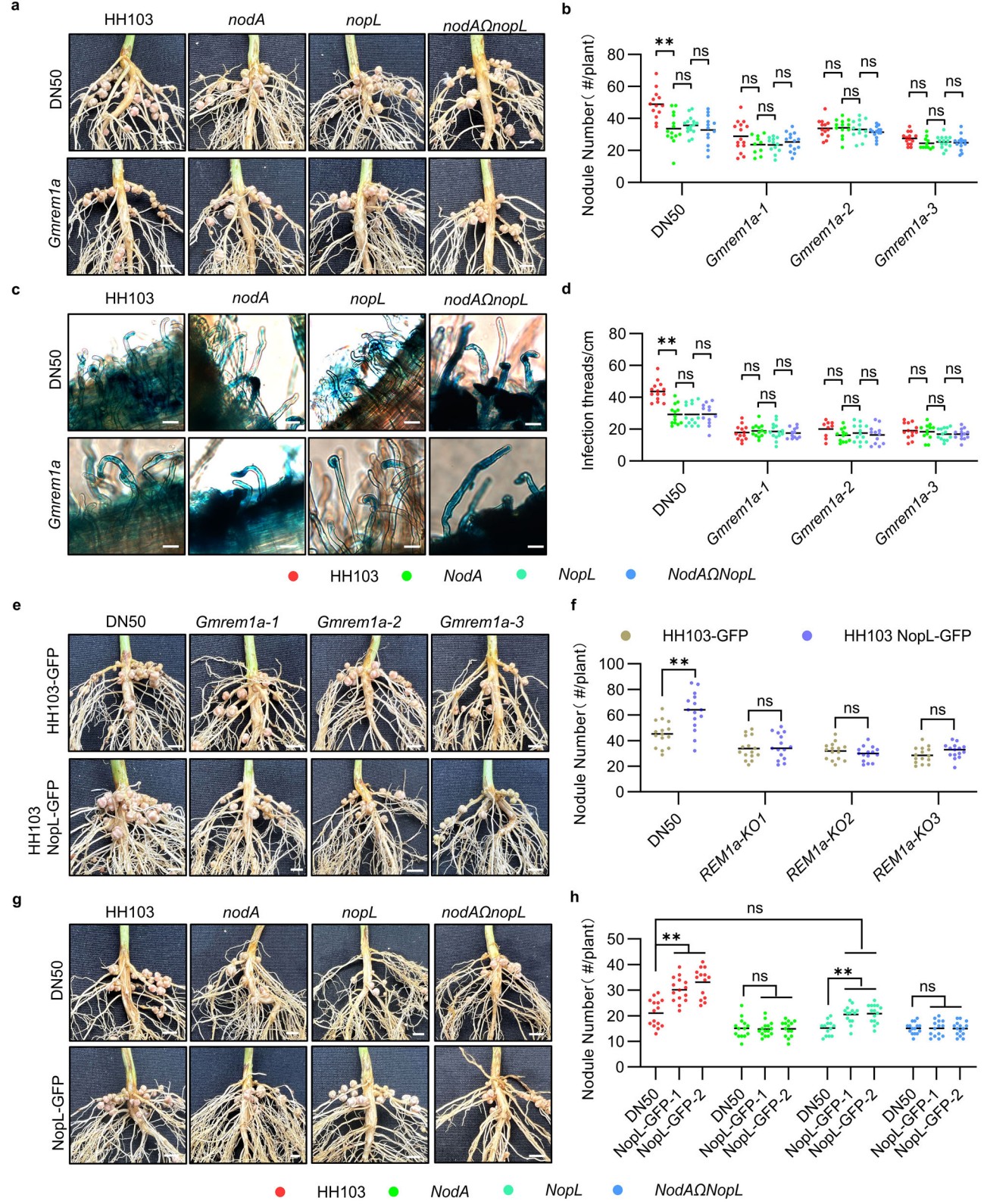

**Fig. 4 | GmREM1a is required for NopL action in DN50. a** Roots of wild type (DN50) or *Gmrem1a* soybean plants 28 days post inoculation (dpi) with HH103, *nodA*, *nopL* and *nodAΩnopL* mutants. Scale bars = 4 mm. **b** Nodule quantification per plant for (**a**). **c** Infection threads formation in roots of DN50 and *Gmrem1a* plants at 1 dpi. Scale bars = 20 μm. **d** Number of infection threads (per cm of root) for (**c**). **e** Roots of DN50 and *Gmrem1a* alleles inoculated with HH103 tagged with GFP (HH103-GFP) or HH103 overexpressing NopL (HH103(NopL-GFP)) 28 days post inoculation (dpi). **f** Nodule number per plant for (**e**). Scale bars = 4 mm. **g** Roots of DN50 and *NopL-GFP* transgenic plants inoculated with HH103, *nodA*, *nopL,* and *nodAΩnopL* mutants 28 days post inoculation (dpi). Scale bars = 4 mm. **h** Nodule number per plant for (**g**). Statistical analysis used two-sided Student's *t*-test (** for *P* < 0.01 and ns not significant). n≥ 15, three biological replicates. Source data are provided as a Source Data file.

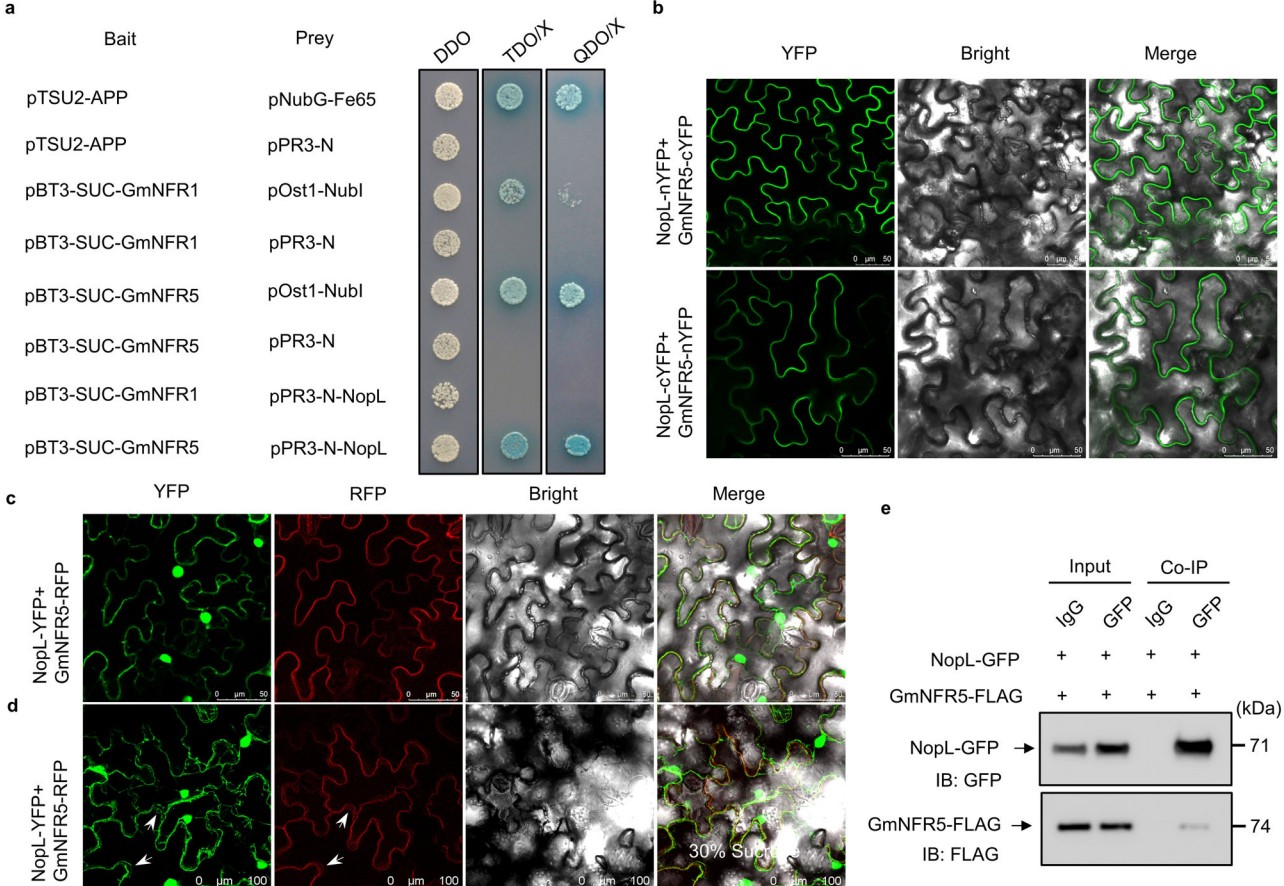

**Fig. 5 | NopL interacts with the MtNFR5 ortholog GmNFR5. a** Membrane Yeast Two Hybrid (MbY2H) assay showing NopL interaction with GmNFR5 but not with GmNFR1a. **b** BiFC analysis of the interactions between NopL and GmNFR5. In the top and down panels, the split YFP is inversely fused in C or N position for the two proteins. Scale bars = 50 μm. **c** BiFC analysis showing the co-localization of NopL and GmNFR5 in *N. benthamiana* cells. Scale bars = 50 μm. **d** BiFC analysis for co-localization of NopL and GmNFR5 following plasmolysis with 30% sucrose. Scale bars = 50 μm. **e** Co-IP assay showing that NopL interact with GmNFR5 in soybean cells. kDa kiloDaldons, IB: FLAG Imunoblot using FLAG AB, IB: GFP Imunoblot using GFP AB. The experiments in b-e were repeated three times with similar results. Source data are provided as a Source Data file.

To further investigate the role of NopL on NF signaling, we constructed stable transgenic soybean plants overexpressing *NopL* expressed from the *GmREM1a* promoter in the DN50 background (named NopL-GFP; Supplementary Fig. 8f). NopL-GFP expressing plants inoculated with HH103 or *nopL* mutant strains produced more nodules than the wild-type (Fig. 4g, h). Interestingly, NopL-GFP plants inoculated with the *nopL* mutant strain produced similar nodule number as the DN50 plants inoculated with HH103, showing that the *nopL* mutation in the bacteria can be complemented by *NopL* expression in *planta* (Fig. 4g, h). In contrast, the number of nodules was not increased when these plants were inoculated with *nodA* and *nodAΩ-nopL* mutants (Fig. 4g, h) implying that NF is required for NopL action. In addition, we overexpressed *NopL* under the control of *GmREM1a* promoter by hairy root transformation in *Gmrem1a* mutant plants. Overexpression of *NopL* did not increase the number of nodules inoculated with HH103 or the *nodA, nopL* and *nodAΩnopL* mutants in *Gmrem1a* (Supplementary Fig. 9d–f). In conclusion, these results indicate that GmREM1a acts as a positive regulator of nodule formation, and that both NF signaling and NopL action depend on GmREM1a.

## GmREM1a interacts with GmNFR1 and GmNFR5
As MtSymREM1 interacts with the NF receptors NFP and LYK3[26,40], we tested GmREM1a interaction with GmNFR1 and GmNFR5. BiFC analysis indicated that GmREM1a interacts with GmNFR1 or GmNFR5 in *N. benthamiana* epidermal cells (Supplementary Fig. 10a). To further

confirmed GmREM1a interactions with GmNFR1 or GmNFR5 using the yeast-based DUAL membrane system[47] (MbY2H; Supplementary Fig. 10b). Furthermore, Co-IP experiments using hairy root tissues expressing GmNFR1-FLAG or GmNFR5-FLAG also showed that GmREM1a interacts with GmNFR1 and GmNFR5 in vivo (Supplementary Fig. 10c). These results support that GmREM1a interacts with GmNFR1 and GmNFR5 in soybean root tissue.

## NopL physically interacts with GmNFR5
Next, we tested if NopL can interact with GmNFR1 and GmNFR5 using the yeast two-hybrid DUAL membrane system. NopL interacts with GmNFR5 but not with GmNFR1a in yeast cells (Fig. 5a). BiFC experiments in *N. benthamiana* cells also showed that NopL interacts with GmNFR5 (Fig. 5b). In agreement with the membrane localization seen previously, NopL-YFP and GmNFR5-RFP protein fusions expressed in *N. benthamiana* cells co-localized to the membrane (Fig. 5c, d). Furthermore, Co-IP experiments using hairy root tissues expressing *GmNFR5-FLAG* and *NopL-GFP* also showed that NopL interacts with GmNFR5 in soybean in vivo (Fig. 5e). These results indicate that NopL can physically interacts with GmNFR5.

## NopL promotes the GmREM1a-GmNFR5 interaction
During early stages of the rhizobium legume interaction, MtSymREM1 is known to act as a scaffold protein to recruit the symbiotic receptors in membrane nanodomains to perceive NF[26]. As western blot analysis

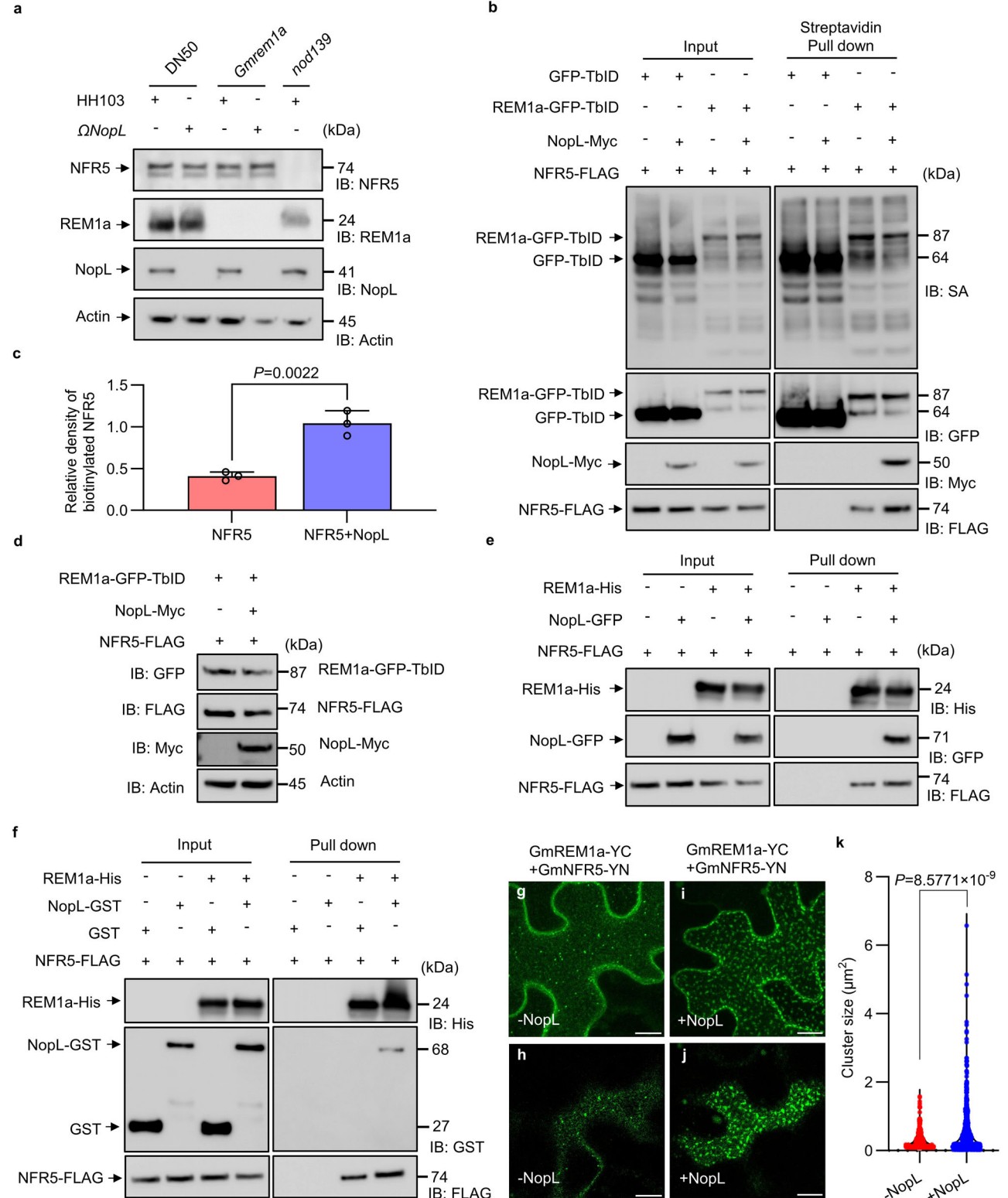

showed that GmREM1a, GmNFR5 and NopL were present in soybean root hair cells (Fig. 6a), we tested whether NopL would help GmREM1a recruitment of GmNFR5. To do this, a construct with a constitutively expressing GmREM1a fused to GFP and TurboID (TbID) was generated (*pGmREM1a: GmREM1a-GFP-TbID*) in order to biotinylate GmREM1a associated proteins[48,49]. Subcellular localization analysis confirmed that the fusion of GmREM1a to TbID did not affect its subcellular localization (Supplementary Fig. 11a). We then co-expressed *pGmREM1a: GmREM1a-GFP-TbID* and *GmNFR5-FLAG* with or without

NopL-Myc in soybean hairy roots. GFP-TbID was co-expressed with GmNFR5-FLAG as a control. After pull down by streptavidin agarose from the protein extracts, immunoblotting with anti-FLAG antibody detected GmNFR5-FLAG and anti-GFP antibody detected GmREM1a-GFP-TbID. Proximity labeling analysis showed that the co-expression of the NopL protein promoted GmREM1a-GFP-TbID labeling of GmNFR5 in soybean root cells (Fig. 6b, c) without significantly increasing the GmREM1a-GFP-TbID and GmNFR5-FLAG protein content (Fig. 6d). To support this finding, we performed a similar

**Fig. 6 | NopL promotes the interaction of GmREM1a to GmNFR5. a** Detection of GmREM1a, NopL and GmNFR5 in soybean hairy roots extracts. Actin was used as the loading control. DN50 inoculated with HH103 was the positive control, while the *nopL* mutant was the negative control. **b** Proximity Labeling (PL) assay for biotinylation of NFR5 in soybean hairy roots. GFP-TbID or REM1a-GFP-TbID fusion proteins were co-expressed with GmNFR5-FLAG in the presence or absence of NopL-Myc. Immunoblotting (IB) antibodies are indicated on the left, and detected proteins (kDa) on the right. **c** Relative amount of biotinylated GmNFR5 detected in (**b**). Gray analysis of the protein content of biotinylated GmNFR5 after Streptavidin Pull down. The ratio of the gray value of GmNFR5 versus GmREM1a detected in the experiment shown in (**b**) was used to calculate the relative density of biotinylated GmNFR5. Data are represented as mean ± SD, and statistical analysis used Student's *t*-test (two-sided). Three biological replicates were performed. **d** Detection of REM1a-GFP-TbID, NopL-Myc and NFR5-FLAG in soybean hairy roots from (**b**). Actin

was the loading control. **e** Semi-pull-down assay showing that the interaction of NopL with GmREM1a in vitro promotes GmREM1a-His to pull down more GmNFR5 from protein extracts of *3SS: GmNFR5-FLAG* hairy roots. **f** Interaction of NopL with GmNFR5 in vivo promotes GmREM1-His to pull down more GmNFR5 from protein extracts of *3SS: GmNFR5-FLAG* and *pREM1a: NopL-GFP* hairy roots. Confocal microscopy images maximal projection (**g**) and of surface (**h**) view of *N. benthamiana* expressing *GmREM1a-YC* and *GmNFR5-YN* (-NopL); Confocal microscopy images maximal projection (**i**) and of surface (**j**) view of *N. benthamiana* expressing *GmREM1a-YC* and *GmNFR5-YN* plus *NopL* (+NopL). Scale bar = 10 μm.
**k** Quantification of the size of the GmREM1a-YC /GmNFR5-YN clusters with and without NopL. Statistical analysis used Student's *t*-test (two-sided). IB: FLAG imunoblot using FLAG AB, IB: GFP imunoblot using GFP AB, IB:His imunoblot using His AB, IB:GST imunoblot using GST AB. The experiments were repeated three times with similar results. Source data are provided as a Source Data file.

experiment using a *pGmREM1a:NopL-GFP-TbID* construct co-expressed with a *GmNFR5-FLAG* construct in presence or not of a *3SS:GmREM1a-myc* construct. This experiment showed that GmREM1a promotes NopL-GFP-TbID labeling of GmNFR5 in soybean root cells (Supplementary Fig. 11b–e). In addition, semi-pull-down analyses showed that interactions between NopL and GmREM1a in vitro, or NopL and GmNFR5 in vivo promote GmREM1a to pull down more GmNFR5 (Fig. 6e, f). To further validate whether NopL facilitates the recruitment of GmNFR5 to nanodomains by GmREM1a, we performed BiFC assay of GmREM1a and GmNFR5 in the presence or absence of NopL and visualized them by confocal microscopy. This analysis shows that NopL indeed promotes GmREM1a interaction with GmNFR5 and increases GmREM1a-YC/GmNFR5-YN cluster sizes in *N. benthamiana* (Fig. 6g–k). These results suggest that NopL is interaction with GmREM1a and GmNFR5, that enhances GmREM1a and GmNFR5 interaction and promotes GmREM1a to recruit GmNFR5 to the nanodomain.

## GmREM1a and NopL modulate soybean symbiotic gene expression

To further determine the effect of NopL on NFs signaling, we performed the RNA sequencing (RNA-Seq) analysis of DN50 plants inoculated with the *nodA* or *nopL* mutants. 2658 and 2006 genes were differentially expressed in DN50 inoculated with *nodA* or *nopL* mutants compared to HH103 (Supplementary Data 3 and 4). Furthermore, 554 of differentially expressed genes (DEGs) were common to *nodA* and *nopL* mutants (Supplementary Fig. 12a) and their expression showed a clear positive correlation (*R*, Pearson correlation coefficient = 0.8; $p < 2.2 \times 10^{-16}$; Supplementary Fig. 12b). Consistent with their phenotype, a number of symbiotic-genes, including *NIN*, had similar expression patterns in the plants inoculated with the *nodA* or *nopL* mutants (Supplementary Fig. 12c). These results further support that NopL participate to NF mediated signaling.

The above results showed that the NopL action depends on GmREM1a during NF signaling. In order to further show the wider impact of GmREM1a interaction with the T3E NopL in NF signaling, RNA-seq experiments were undertaken using DN50 plants and *Gmrem1a* mutant nodulated by HH103 or *nopL* mutant rhizobial strains. In the absence of rhizobial inoculation the basal level of expression of many symbiotic genes, including *NINa*, *NSP1s*, *CYCLOPS*, and *NFRs* was significantly reduced in the *Gmrem1a* mutant (Fig. 7a). Rhizobial inoculation of WT plants activated genes of the common symbiotic signaling pathway (CSSP) including *NINa*, *NSP1s*, *CYCLOPS*, and *CCaMK* but these were all mis-expressed in the *Gmrem1a* mutant background compared to the control. This altered expression of genes from the CSSP in the *Gmrem1a* mutant was observed irrespective of the inoculation with either HH103 or the *nopL* mutant (Fig. 7a). Wider assessments identified 596 differentially expressed genes (DEGs) in DN50 and *Gmrem1a* plants when inoculated with HH103, and these include down-regulated symbiotic key genes (Supplementary Fig. 13a).

In agreement with the hypothesis that both GmREM1a and NopL function in the common signaling pathways, there were 83 common DEGs between the *nopL* and *Gmrem1a* mutants (Fig. 7b, Supplementary Data 3 and 5) and their expression showed a positive correlation ($R = 0.68$; $p < 2.2 \times 10^{-16}$; Fig. 7c). More importantly, the expression of common DEGs in the *Gmrem1a* mutant inoculated with HH103 (WT) or *nopL* (mutant) compared to DN50 inoculated with HH103 showed a clearer positive correlation ($R = 0.98$; $p < 2.2 \times 10^{-16}$; Fig. 7d; Supplementary Data 6). This result confirms that the function of NopL depends on GmREM1a.

Within the 83 DEGs, a homolog of *LjRINRK1* (*Lotus japonicus* Rhizobial Infection Receptor-like Kinase1), a homolog of *MtPLT1* (*Medicago truncatula* PLETHORA 1) and *GmmiR172c* were identified, and both *GmRINRK1* and *GmmiR172c* were downregulated in DN50 inoculated with the *nodA* mutant (Supplementary Fig. 12c, Supplementary Fig. 13a). *LjRINRK1*, *MtPLT1*, and *GmmiR172c* are key factors that coordinates the output of NF signaling required for rhizobial plant infections[50–52] or nodule development[53]. Given, their importance, we examined the expression of *GmRINRK1*, *GmPLT1* and *GmmiR172c* in WT and in the *Gmrem1a* backgrounds using qRT-PCR. After inoculation with either the *nodA* mutant, *nopL* mutant or *nodAΩnopL* double mutant, *GmRINRK1*, *GmPLT1* and *GmmiR172c* expression in DN50 was significantly lower than when inoculated with HH103 (Fig. 7d, Supplementary Fig. 13b, c). In the *Gmrem1a* mutant, *GmRINRK1* expression was low when inoculated with HH103 and the mutant derivatives (Fig. 7e). The *GmPLT1* and *GmmiR172c* genes also showed altered expression, but this could not be associated with the nodule phenotype of *Gmrem1a* inoculated with HH103 and its mutant (Supplementary Fig. 13b, c). This shows that *GmRINRK1* expression strictly depends on a functional GmREM1a and on the NF signaling, whereas *GmPLT1* and *GmmiR172c* expressions are not. The expression of important other symbiosis-related genes was also explored. Expression of *GmNINa* was reduced in the rhizobial mutant backgrounds compared to WT but these patterns were not significantly changed in the *Gmrem1a* mutant (Fig. 7f). This suggested that *GmNINa* expression is independent of GmREM1a but requires NopL, as well as the NF signaling. The relative expression of *GmNFR5* was significantly increased in the *Gmrem1a* mutant (Fig. 7g) suggesting an inhibitory action of GmREM1a on *GmNFR5* expression.

## Discussion

In the legume rhizobia symbiosis, a conserved NF signaling pathway allows the recognition of the NF produced by rhizobia in order to establish symbiosis[54,55]. In addition, T3SS play multiple roles in regulating host specificity, immunity, nodule senescence or even allowing NF recognition to be by-passed[31,42,56]. Our study reveals that NF-mediated symbiosis in soybean requires GmREM1a and NopL. It suggests a mechanistic model (Supplementary Fig. 14) in which NF-mediated symbiosis and T3E-mediated secretion can affect NF signaling in soybean, with NopL playing an essential role in which the

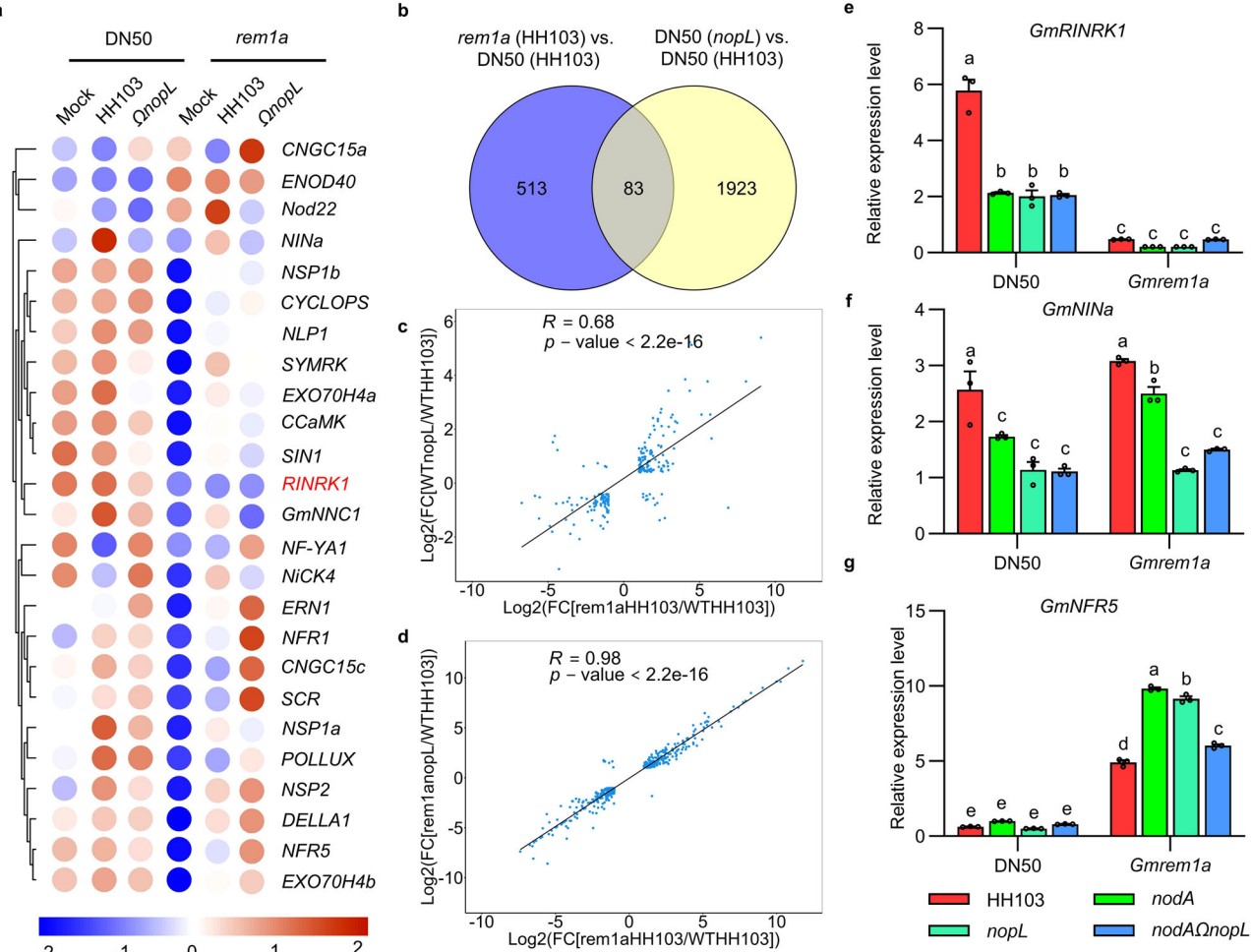

**Fig. 7 | RNA-seq analysis of roots of WT and *rem1a* inoculated with HH103 or *nopL* mutant at 24 dpi. a** Heat maps of symbiosis-associated genes in WT or *rem1a* mutants inoculated with MgSO$_4$ (Mock), HH103 or *nopL* mutants. The scale bar at the bottom of the panel indicates the log$_2$FPKM (Fragments Per Kilobase Million). **b** Venn diagrams showing unique and common genes amongst differentially expressed genes (DEGs) for different pairwise comparisons. The data of *Gmrem1a* (HH103) vs. DN50 (HH103) set is shown in blue, and the data of DN50 (*nopL*) vs. DN50 (HH103) set is shown in yellow. **c** Scatterplot showing the expression correlation of DEGs in DN50 inoculated with *nopL* mutants and in *Gmrem1a* plants inoculated with HH103 compared with DN50 inoculated with HH103. **d** Scatterplot showing the expression correlation of DEGs in *Gmrem1a* plants inoculated with HH103 or *nopL* mutants compared with DN50 inoculated with HH103. The black line is the linear regression. *R*, Pearson correlation coefficient. FC, Fold change in **c**, **d**. **e**–**g**, Relative expression level of *GmRINRK*, *GmNINa* and *GmNFR5* in roots of WT or *rem1a* mutants inoculated with HH103, *nodA* mutants, *nopL* mutants or *nodA* and *nopL* double mutant in 24 dpi. Values are means ± SD (*n* = 3 biological repeats) and *P* < 0.05 by ANOVA with Tukey's multiple comparison tests. Source data are provided as a Source Data file.

**NopL effector affects NF-mediated symbiosis signaling.** NopL promotes the interaction between GmNFR5 and GmREM1a showing that rhizobial T3E not only hijack the immune system of their symbiotic hosts[32,42], but can also play a direct role in NF signaling. T3E NopL is broadly conserved in rhizobia that nodulate soybean, and a NF plus T3E dependent nodulation pathway also exists[42]. In agreement with the role of NopL in NF signaling, *nopL* mutants did not exhibit a change in nodule number of the NF-independent strain *Bradyrhizobium sp.* ORS3257 nodulating *Aeschynomene indica*[32,57]. However, in the NF-dependent symbiosis of *Vigna mungo*, the *nopL* mutant or *nodC* mutant of *Bradyrhizobium elkanii* USDA61 exhibited a similar nodulation minus phenotype[58]. These results are consistent with the model (Supplementary Fig. 14). *Sinorhizobium meliloti* does not have a type III secretion system, so NopL does not exist in the *S. meliloti* 1021 or *Ensifer medicae* WSM419 strains, the symbiotic partners of *M. truncatula which*[59,60], this regulation by T3E might be absent in this symbiotic system or other factors might play similar roles in this plant. Since NopL is very conserved in soybean and some other legume rhizobia, this model may exist in many legume-rhizobia interactions.

An intriguing observation of our work is that soybean-rhizobia symbiosis is partly NF independent, but this agrees with some previous studies[42]. *nodA*, *nopL*, *nodAΩnopL* mutants induced fewer nodules than the wild-type HH103 strain, indicating that NF and NopL both play positive but also non-essential roles during nodule formation of SN14 and DN50, suggesting a NF independent nodulation in some soybean varieties[43].

NopL is a T3E specific to rhizobia[31] secreted into the host cell by the rhizobial T3SS to physically interact with GmREM1a and GmNFR5. Although the *Gmrem1a* mutants blocks the interaction between NF signaling and NopL, nodules are still formed on the mutant plants, in agreement with previous studies[24,26,61]. The similar phenotype seen in the *Gmrem1a* mutant and the bacterial *nodA* and *nopL* mutants further supports the hypothesis that they participate in the same signaling pathway. Following GmREM1a / GmNFR5/ NopL interaction our results indicate that *GmNINa* and *GmRINRK1* expression are activated. Normal *LjRINRK1* expression is regulated by NIN in *L. japonicus*[50]. However, the low expression of *GmRINRK1* in the *Gmrem1a* mutant indicates that there might exist other GmNINa-independent regulation mechanism for *GmRINRK1* expression. This might also explain the NF-independent

symbiosis observed in the *Gmrem1a* mutant. As the *nopL* mutant induces fewer ITs and primordia compared to the wild-type rhizobium, this suggests that GmRINRK1 participate to rhizobial infection but not to nodule development[50].

GmREM1a is a pivotal protein in the recruitment of GmNFR5 and GmNFR1 for NF recognition. It also stabilizes the NF receptors at the membrane during symbiotic establishment[24,26]. This function might be explained by remorin induced alterations in membrane fluidity[24,62,63], higher order protein oligomerization[64–66], maintenance of membrane-associated and phase-separated condensates[67] or influencing membrane bending and stabilization[24]. The role of NopL in these functions should be addressed in future research.

In parallel with Nod factor-dependent partner selection, the plant immune system also plays an important role in influencing the establishment of symbiosis[68,69]. In tobacco, NopL of the *Sinorhizobium* strain NGR234 was phosphorylated by the mitogen-activated protein kinase NtSIPK, which is often linked to plant defense[40,41]. We previously showed that NopL plays a negative role on the nodule formation of the soybean landrace Qingdou[70], suggesting that in addition to its promotion of the NF signaling, NopL might trigger effector-triggered immunity in different soybean genotypes, and/or the interactions with the NF signaling cascade may differ in certain genotypes. NopL can also interfere with nodule functioning by controlling senescence as observed in *Phaseolus vulgaris*[41], but this aspect was not studied yet in soybean. It would be interesting to know if this action is also NF dependent by testing the *nodA* and *nodAΩnopL* in this system. NopL could also play a role in the subtle balance between immunity and senescence during symbiosis depending on the legume genotypes[71]. More studies will be required to know if GmREM1a, GmNFR5, and GmNFR1 alleles are related to the NopL type (positive/negative) of action in these different landraces.

In summary, our study shed light on a long-standing question in the soybean symbiotic interaction and contributes to our understanding of how NF signaling pathways and the T3E interact to affect early stages of soybean symbiosis. Our results also explain why symbiotic interactions in some soybean varieties depend on complementary NF signaling or T3E events[42]. These results may have broad practical implications, as they suggest that controlling NopL expression could affect NF signaling and consequently the effectiveness of the symbiotic nitrogen fixation in soybean.

## Methods

### Plant materials, rhizobia, and growth conditions

The Suinong 14 (SN14) and Donong 50 (DN50) cultivar were used throughout the study. The DN50 was used in the generation of all stable transgenic and gene-edited soybean genotypes, *Gmrem1a*-knock out lines, *Gmrem1a*-knockdown lines.

For the nodulation tests, soybean seeds were sterilized overnight with chlorine gas and then planted in plastic jars. The soybean seedlings were grown in a greenhouse under a 16 h light/8 h dark cycle at a temperature of 25 °C. They were inoculated with *S. fredii* HH103 or mutants in this background. The soybean seedlings were inoculated with rhizobia suspended in sterile water containing 10 mM MgSO$_4$ at OD$_{600}$ = 0.2. The nodulation phenotype of soybean was recorded 28 days post-inoculation (dpi) with rhizobia. The soybean seedlings were provided with a 1 mM N solution.

*S. fredii* HH103 and its mutants (*nodA* mutant, *nopL* mutant[70], *ttsI* mutant[72], *nodQ* mutant, *nodAΩnopL* mutant, *nodQΩnopL* mutant, *nodB* mutant[73], *nodC* mutant, *nodBΩNopL* mutant and *nodCΩnopL* mutant), as well as GFP or GUS-tagged HH103 were grown at 28 °C in TY medium.

For *nodA* and *nodQ* mutant construction, a kanamycin Ω interposon was ligated into *nodA* or *nodQ* downstream of the ATG codon, then cloned into the suicide vector pJQ200SK[74]. The triparental mating was used to transfer the suicide vector from *Escherichia coli* DH5α cells

into *S. fredii* HH103 in the presence of pRK2013 helper plasmid[75]. The same approach was used to construct *nodAΩnopL* mutant and *nodQΩnopL* mutant on *nodA* mutant or *nodQ* mutant. Mutant strains were verified using western blots or expression studies (Supplementary Fig. 2).

*Agrobacterium* strains, including *Agrobacterium tumefaciens* EHA105 and GV3101 (pSoup-p19) and *Agrobacterium rhizogenes* strain K599, were grown at 28 °C in YEP medium. *E. coli* was cultured at 37 °C in LB medium.

### GUS staining

To observe the early infection events after inoculation with HH103 and its mutants, soybean roots were collected at 1 dpi and 7 dpi, respectively. The collected soybean roots were placed in a GUS staining solution (50 mM sodium phosphate buffer, pH 7.2, 0.5 mM K$_3$Fe (CN)$_6$, 0.5 mM K$_4$Fe (CN)$_6$, 2 mM 5-bromo-4-chloro-3-indolyl-β-D-glucuronic acid, and 0.3% Triton X-100), then kept under vacuum for 10 min, and subsequently incubated for 24 h at 37 °C in the dark. This was followed by five washes of 30 min each using de-staining Buffer 1 (75% [v/v] ethanol and 25% [v/v] acetic acid) and de-staining Buffer 2 (75% [v/v] ethanol and 25% [v/v] distilled water).

### Analysis of rhizobial infection events

The infection events were observed using a fluorescent microscope (Leica, DM2500). Infected segments of 1 cm lateral roots were taken at 1 dpi and the number of infection threads in the root segments was counted, and three independent lateral root segments of each plant were used for each biological replicate. Five biological replicates of infestation events were performed for each GUS-tagged HH103.

For the analysis of the number of nodule primordium soybean roots after GUS staining at 7 dpi. Fifteen biological replicates were performed for each condition.

### Antibody production and verification

The NopL peptide (amino acid residues from 1 to 300), GmREM1a peptide (amino acid residues from 1 to 194) and GmNFR5 peptide (amino acid residues from 211 to 242) were used as the antigen to develop specific antibody by Abmart. To verify the specificity of the NopL antibody, cell pellet of HH103, HH103Ω*nopL*, HH103Ω*ttsI* and HH103 *NptII:NopL:GFP* were collected, added to 2 × SDS-PAGE loading buffer and heated at 100 °C for 10 min, separated on 12% PAGE gels for immunoblot analysis. To verify the specificity of GmREM1a antibodies. Soybean roots of DN50, *Gmrem1a* mutants at 1 dpi, respectively, were collected and ground to a fine powder in liquid nitrogen. Extraction buffer (50 mM Tris-HCl, pH 7.4, 150 mM NaCl, 1% Triton X-100, 0.5 mM EDTA, 1 mM DTT and 1 × protease inhibitor mixture [Roche]) was added to the samples. After homogenization on ice for 30 min, the mixture was centrifuged at 15,000 × *g* for 20 min at 4 °C. The extracts were heated at 100 °C for 10 min with 2 × SDS-PAGE loading buffer and separated on 12% PAGE gels for immunoblot analysis. To verify the specificity of GmNFR5 antibodies, protein extracts of DN50 roots were assayed by immunoblotting and *nod139*[76] (a mutant of NFR5 in soybean cultivar Bragg) was used as a negative control.

### Immunohistochemistry and immunolocalization

Immunohistochemistry and immunofluorescence were performed following previously described methods[77,78]. Soybean root nodules, collected at 10 dpi, were fixed in Formaldehyde-ethanol-acetic acid fixative solution (FAA fixative Solution), dehydrated with ethanol, and embedded with wax. The material was sectioned to 8 µm and dewaxed on slides for antigen repair as well as BSA blocking and incubated overnight at 4 °C with the corresponding antibody (anti-NopL, 1:150 dilution. anti-REM1, 1:200 dilution). After the incubation, the slides were washed five times with PBS and then incubated with HRP-conjugated goat anti-Rabbit IgG (Cy3-conjugated goat anti-Rabbit IgG

(Abbkine, A22220, 1:500 dilution) or FITC-conjugated goat anti-Rabbit IgG (Abbkine, A22120, 1:500 dilution) for immunofluorescence, for 1 h at 4 °C. After washing the slides with PBS, the root nodule sections were incubated with hematoxylin or DAPI solution for 20 min. Immunofluorescence was preserved using antifade medium to maintain fluorescence before imaging with a confocal microscope.

### Electron microscopy

For immunoelectron microscopy, nodule tissue was collected 10 days after inoculation. The collected nodule tissues were fixed in 2.5% glutaraldehyde fixative (Servicebio, G1124) and then subjected to cryo-dehydrated permeabilized LR White resin (Sigma-Aldrich, L9774) cryo-polymerized. Subsequently, sections were incubated with anti-NopL IgG (1:5 dilution) and immunolabeled using goat antibody against rabbit IgG conjugated with 10 nm (Sigma-Aldrich, G7402) diameter gold particles[79].

### Cya reporter translocation assays

NopL was fused to the calmodulin-dependent adenylate-cyclase (Cya) domain and expressed from the *NopL* promoter in HH103 or *ttsI* mutant. To analyze that NopL can translocate in soybean root cells, detect cAMP content in soybean roots inoculated with HH103 *NopL-Cya* (Cyclic AMP ELISA Kit, Cayman, 481002) at 1 dpi. The soybean roots inoculated with HH103 and *ttsI NopL-Cya* as negative controls.

### Interacting protein identification of NopL by LC−MS/MS

The coding sequence of the *NopL* was cloned into pET28a vector using *Bam*H I and *Sal* I restriction enzymes. The resulting vector was transformed into *E. coli* BL21 (DE3) for the expression of recombinant protein. His-NopL protein was purified by Ni NTA Beads 6FF (Smart-lifesciences, SA005005). DN50 roots inoculated with HH103 were collected at 1 dpi and frozen in liquid nitrogen. Soybean root hair cells were collected by scraping the soybean roots using the cell scraper (Corning Incorporated, 3010). Soybean root hair proteins were extracted using protein extraction buffer (50 mM Tris-HCl pH 7.5, 150 mM NaCl, 10% Glycerol, 0.2% Trixton X-100, 1 mM DTT, 1×protease inhibitor cocktail, 1 mM NaF, 1 mM $Na_3VO_4$ and 1 mM PMSF). The obtained His-NopL protein was added to the soybean root protein extract, while no His-NopL protein was added to the control sample. After mixing, the NopL interacting proteins were pull-down using Ni NTA Magarose Beads (Smart-lifesciences, SM008001) semi-in vivo. Finally, the eluted proteins were subjected to analysis using LC-MS/MS to identify and characterize the proteins that interact with NopL.

For LC-MS/MS analysis, trypsin-digested peptides were analyzed by an EASY-nLC 1200 system (Thermo, USA) coupled with a Q Exactive HF-X quadrupole orbitrap mass spectrometer (Thermo, USA). The peptide identification was performed by PEAKS Studio 8.5 software (https://www.bioinfor.com/peaks-85-release/). The parameters were set as follows: precursor mass tolorance is 10 ppm; fragment mass tolorance is 0.05 Da. False discovery rate (FDR) of peptide identification was set as FDR ≤ 0.01. A minimum of one unique peptide identification was used to support protein identification.

### BiFC (bimolecular fluorescence complementation) assays

The BiFC assays were conducted following previously established protocols[80]. *A. tumefaciens* GV3101 (pSoup-p19) carrying different combinations of cYFP (C-terminal yellow fluorescent protein) or candidate genes fused to cYFP or nYFP (N-terminal yellow fluorescent protein) constructs were resuspended in permeabilization buffer (10 mM MES adjust pH to 5.7 used KOH, 10 mM $MgCl_2$, 150 µM acetosyringone) at an optical density of $OD_{600} = 0.2$, gently mixed and permeabilized into young leaves of expanded *N. benthamiana*. Fluorescence signals were observed after 48 h using laser confocal microscopy (Leica, TCS SP8).

For the BiFC analysis of the interactions between GmREM1a and GmNFR5 with or without NopL, *A. tumefaciens* (GV3101) carrying the plasmid of interest was grown in YEP liquid culture overnight at 28 °C with the corresponding antibiotics. The liquid culture was centrifuged (3200 × *g*, 10 min) and washed twice with 10 mM $MgCl_2$. Finally, the bacteria were re-suspended in 10 mM $MgCl_2$ solution ($OD_{600} = 0.4$) and mixed with p19 ($OD_{600} = 0.1$) in the presence of 200 µM acetosyringone, and then incubated in the dark for 2 h at room temperature before infiltration into *N. benthamiana* leaves. After 2 days of infiltration, images were taken using confocal laser-scanning Microscopy (Leica TCS SP8 confocal microscope equipped with 63x water immersion lenses, YFP was excited at 514 nm using an argon laser and detected at 525–560 nm).

### Co-immunoprecipitation

For the Co-immunoprecipitation analysis of the interaction between NopL and REM1a/1b, *A. rhizogenes* K599 carrying a construct allowing the co-expression of *35S:NopL-GFP* and *35 S:REMa/b-Myc* was used to produce hairy roots expressing *NopL-GFP* and *REM1a/1b-Myc* in soybean. After agrobacterial transformation, roots of soybean were collected, crushed in liquid nitrogen, and total protein was extracted using protein extraction buffer (50 mM HEPES pH 7.5, 150 mM NaCl, 10% Glycerol, 0.2% Trixton X-100, 1 mM DTT, 1×protease inhibitor cocktail and 50 µM MG132). The total protein extract was then transferred to two new microtube (left 1% as input) and 20 µL of ChIP magnetic A beads (Sigma-Aldrich, 16-661) were added. after mixing, anti-GFP antibody (MBL, 598, used at 1:500 dilution) or Rabbit IgG (Sigma-Aldrich, NI01, used at 1:500 dilution) were added separately and incubated for 3-4 h at 4 °C on a roller shaker. For immunoblotting assays, anti-GFP antibody (Abmart, M20004, used at 1:2000 dilution) or anti-Myc antibody (Abmart, M20002, used at 1:2500 dilution) was used to detect NopL-GFP or REMa/b-Myc, respectively.

For the in vivo interaction detection of GmREM1a with GmNFR1 or GmNFR5, *35S: GmNFR1-FLAG* or *35S: GmNFR5-FLAG* were introduced into *A. tumefaciens* K599 and used for soybean hairy root transformation. Root proteins were extracted and incubated with FLAG beads (Sigma-Aldrich, M8823) for 4 h at 4 °C. Beads were washed 6 times with PBS. For immunoblotting assays, anti-FLAG antibody (Invitrogen, MA1-91878, used at 1:10,000 dilution) or anti-REM1a antibody (used at 1:2500 dilution) was used to detect NFR1/5-FLAG or REMa, respectively.

### GST pull-down

His-GmREM1a was incubated with glutathione GST or GST-NopL for 2 h in an interaction buffer (20 mM Tris-HCl, 100 mM NaCl, 0.1 mM EDTA, and 0.2% Triton X-100, pH 7.4). After incubation, Glutathione Magarose Beads (Smart-lifesciences, SM002001) were added to the reaction mixture, and the reaction mixture (left 10% as input) was incubated for 3-4 h at 4 °C. Following this incubation, the beads were washed with PBS to remove non-specific binding. The proteins bound to the beads were eluted by added 100 µL of 2×protein loading buffer and heating at 100 °C for 10 min. The target proteins were detected used anti-GST antibody (Invitrogen, MA4-004, used at 1:2500 dilution) or anti-His antibody (Abmart, M30111, used at 1:2500 dilution).

### Semi-pull-down analysis

For semi-pull-down analysis, NopL-GST or GST proteins were co-incubated with GmREM1a-His proteins and Ni NTA Magarose Beads for 4 h. Soybean hairy root protein extracts containing GmNFR5-FLAG protein were co-incubated for 4 h with preincubated GmREM1a-His Magarose Beads as described above. Pre-incubated Ni NTA Magarose Beads were used as a negative control. After the incubation, the beads were washed 6 times with PBS. The target proteins were detected using anti-GST antibody (Invitrogen, MA4-004, used at 1:2500 dilution), anti-

FLAG antibody (Invitrogen, MA1-91878, used at 1:10,000 dilution) or anti-His antibody (Abmart, M30111, used at 1:2500 dilution).

In additions, GmNFR5-FLAG was overexpressed by hairy roots transformation in transgenic soybean plants expressing NopL-GFP. DN50 containing GmNFR5-FLAG protein was used as a control by hairy roots transformation. Total proteins from transgenic roots inoculated with HH103 at 7 dpi containing NopL-GFP and GmNFR5-FLAG were extracted for semi-pull-down analysis.

## Yeast two-hybrid assays

Yeast two-hybrid assays followed previous methods[81]. Appropriate constructs containing paired genes were co-transformed into strain AH109 using the lithium acetate/carrier DNA/PEG transformation method. pGBKT7-lam/pGADT7-largeT was used as a negative control pair and pGBKT7-p53/pGADT7-largeT was used as a negative control pair. pGBKT7-Lam encodes the Gal4p BD fused with Lamin. pGBKT7-p53 encodes the Gal4p BD fused with murine p53. pGADT7-largeT encodes the Gal4 AD fused with the SV40 large T-antigen.

The yeast split-ubiquitin system followed established protocols[47,82]. Appropriate constructs containing paired genes were co-transformed into strain NMY51 using the lithium acetate/ carrier DNA/PEG transformation method. Ten microliters of yeast suspension was grown on DDO medium plates (SD/-Leu/-Trp), TDO/X medium plates (SD/-Leu/-Trp/-His/X-a-gal) and QDO/X media plates (SD/-Leu/-Trp/-His/-Ade/X-a-gal) (Clontech). The positive control was obtained by co-transfecting pTSU2-APP (positive bait plasmid. APP, amyloid A4 precursor protein) and pNubG-Fe65 (positive prey plasmid. Fe65, amyloid beta A4 precursor protein-binding family B member 1) into strain NMY51. Co-transfection of PBT3-N bait (bait plasmid) and pOst1-NubI (prey plasmid) into NMY51 yeast host strain for functional validation. Ost1 is a resident endoplasmic reticulum protein, ensuring that the Ost1-NubI fusion protein is located in the cytoplasmic region near the cell membrane; NubI is the domain of the wild-type ubiquitin protein, which can actively attract the Cub structure expressed by bait plasmids.

## Soybean hairy root transformation

The soybean hairy root transformation has been previously described[83]. Briefly, *pGmREM1a:GFP* and *pGmREM1a: GmREM1a-GFP* were introduced into *A. tumefaciens* K599, respectively. K599 was incubated in YEP medium containing kanamycin resistance until $OD_{600} = 0.6$. The bacterium was suspended in LCCM (1/10 × Gamborg B5 basal medium, 30 g L$^{-1}$ sucrose, 3.9 g·L$^{-1}$ MES, pH=5.4 and 40 mg L$^{-1}$ acetobutanone). The hypocotyl of germinating soybean seeds was excised and incubated in LCCM containing K599 for 30 min and the surface bacterial solution was aspirated. Seedlings were transferred to CCM medium (1/10 × Gamborg B5 basal medium, 30 g L$^{-1}$ sucrose, 3.9 g L$^{-1}$ MES, pH = 5.4, 40 mg L$^{-1}$ acetobutanone, 400 mg L$^{-1}$ Cysteine and 154.2 mg L$^{-1}$ Dithiothreitol) and incubated for 2 days in dark before being transferred again to rooting medium (1/10 × Gamborg B5 basal medium, 30 g L$^{-1}$ sucrose, 3.9 g L$^{-1}$ MES, pH = 5.4 and 7.5 g L$^{-1}$ agar). Transgenic hairy roots were identified by detecting the RFP marker DsRED, which is encoded by *DsRED2* and is expressed under the control of the *CaMV35S* promoter, adjacent to the target gene inserted in the vector backbone (approximately 1 kb between the *DsRED* and the inserted target gene). RFP was observed used the fluorescent microscope (Leica, MZ10F) and non-positive roots were cut off.

## Affinity purification of biotinylated proteins

*A. rhizogenes* K599 strains carrying *pGmREM1a: GmREM1a-GFP-TbID*, *35 S: GmNFR5-FLAG* or *pGmREM1a: NopL-Myc* were used to co-express *GmREM1a-GFP-TbID* and *GmNFR5-FLAG* or *GmREM1a-GFP-TbID*, *GmNFR5-FLAG*, and *NopL-Myc* by soybean hairy root transformation. Soybean hairy roots were infiltrated with 100 μM biotin and incubated for 2 h in a growth chamber, using *pGmREM1a: GFP-TbID* as a control. Tissue samples were ground in liquid nitrogen and total protein was extracted using protein extraction buffer (50 mM Tris-HCl pH 7.5, 150 mM NaCl, 10% Glycerol, 0.2% Trixton X-100, 1 mM DTT, 1×protease inhibitor cocktail and 50 μM MG132). The supernatants were collected after centrifugation at 15,000 × $g$ for 20 min. To remove free biotin from the total protein, the protein samples were desalted using Zeba™ Spin Desalting Columns (Thermo Fisher Scientific, 89889) according to the manufacturer's instructions, and further concentrated and secondarily desalted using Vivaspin® 500 Centrifugal Concentrator Polyethersulfone (Sartorius, VS0112). Protein samples (500 μg) were incubated with 50 μL of Dynabead C1 Streptavidin beads (Thermo Fisher Scientific, 65001) at 4 °C overnight. The beads were then washed 6 times with protein extraction buffer. Biotinylated proteins were eluted by boiling the beads in SDS sample buffer containing 2 mM of biotin and separated by 10% SDS-PAGE gels.

## RNA-seq and data analysis

For the RNA-seq experiment, two types of soybean plants were used: the DN50 (wild type) and derived *Gmrem1* mutant. These plants were inoculated with different strains of rhizobia, including the wild-type HH103, derived mutants HH103Ω*nopL* and HH103Ω*nodA* and a mock control using 10 mM MgSO$_4$. The plants were inoculated with these rhizobia strains, and the roots were sampled, and total RNA was isolated at 1 dpi. The RNA-seq analysis was performed on the isolated total RNA from the roots of the different soybean plants under the various treatments. Rhizobia inoculation and sample collection were performed as described above. Three roots from different plants were collected as one replicate, and three biological replicates were collected for each treatment. Three individual samples were sequenced using Illumina NovaSeq 6000. To identify differential expression genes (DEGs) between two different samples, the expression level of each transcript was calculated according to the transcripts per million reads (TPM) method. RSEM was used to quantify gene abundances. Essentially, differential expression analysis was performed using the DESeq2. DEGs with |log2FC| ≥ 1 and FDR ≤ 0.05 were considered to be significantly different expressed genes.

## Vectors and primers

All vectors and primers used in this study are shown in Supplementary Data 1.

## Accession numbers

Accession numbers are as follows: *NodA* (*AAY89042.1*), *NopL* (*CEO91525.1*), *GmREM1a* (*Glyma.08G012800*), *GmREM1b* (*Glyma.05G205900*), *GmNFR1* (*Glyma.02G270800*), *GmNFR5* (*Glyma.11G063100*) and *GmUNK1* (*Glyma.12G020500*).

## Statistics and reproducibility

In this study, statistical analysis for this study was done using GraphPad Prism 8.0.1 (GraphPad software http://www.graphpad.com). *P* values less than 0.05 were considered significant and less than 0.01 were considered highly significant. In addition, in all experiments at least three biological replicates were performed in this study. No data were excluded from the analyses and no statistical methods were used to predetermine sample size. For microscopic, confirmation of protein interactions and physiological observations, as least three completely independent experiments were performed to minimize plant-to-plant variations.

## Reporting summary

Further information on research design is available in the Nature Portfolio Reporting Summary linked to this article.

# Data availability

The RNA-Seq data generated in this study have been deposited in the NCBI Sequence Read Archive database under accession code

PRJNA1000775. Processed data have been deposited in the NCBI GEO database under the accession number GSE269425. The protein mass spectrometry of NopL interaction protein raw data used in this study are available in the ProteomeXchange partner repository under accession code PXD052987. Source data are provided with this paper.

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

## Acknowledgements

This work was supported by the National Key Research & Development Program of China (2021YFF1001206, 2023YFD1200600), the National Natural Science Foundation of China (32070274, 32201809, 31771882, 32072014, 590224002 and U20A2027), China Scholar Council fellowships (201906615001). P.R. and K.M. are supported by the CNRS and by a grant (ANR-14-CE19-0003-01) from the French National Research Agency

to P.R. This work has benefited from the support of the LabEx Saclay Plant Sciences (ANR-10-LABX-0040-SPS, LabEx SPS and ANR–17-EUR- 0007, EUR SPS-GSR) which is managed by the French National Research Agency under the program 'Investissements d'avenir' (ANR–11-IDEX-0003-02).

## Author contributions

D.X., P.R., Q.C., Z.T., and C.S. conceived the project. C.M. performed most of the experiments, including the discovery of NopL interaction with GmNFR5 and GmREM1. Y.Y. analyzed the nodulation test. J.W. and Y.G. performed the experiments related to protein–protein interaction. C.M. and H.F. constructed the required vectors. Western blot and genotyping were performed by C.M., J.W., Z.H., C. L., and Y.Y. RNA-Seq analysis was performed by X.D and J.Z. Figures were created by C.M., D.X., and P.R. The manuscript was written by C.M., D.X., L.M., K. M., and P.R. with input from J.W., Y.Y., H.F., M.Y., X.W., Z. Q. and C.L. Funding was acquired by D.X., Q.C., C.L., and J.W. All authors read and approved the final manuscript.

## Competing interests

The authors declare no competing interests.
