## [Peer Review File · Nature Communications]

REVIEWER COMMENTS

Reviewer #1 (Remarks to the Author):

The paper by Ma et al presents interesting data suggesting that the *R. fredii* NopL effector protein is secreted into the soybean plant and interacts with the Nod factor receptor complex to enhance nodulation. Given the role of various effector proteins in plant-pathogen interactions, the model they present is certainly plausible.

However, there are many puzzling aspects of this paper and, in some cases, the data presented do not support their conclusions. Perhaps most puzzling is how they get nodulation of soybean in a *nodA* mutant, which presumably lacks the ability to produce the Nod factor. The logic here seems faulty in that their model depends on the ability of NopL to impact Nod factor recognition and signaling but this is also true with the *nodA* mutant, where presumably there is no Nod factor. Hence, their *nodA* mutant is either quite leaky or something else is strange. I suggest they use a *nodC* mutant and see if they get similar results. Although not discussed extensively, they do mention that (as published by others), in rare cases, soybean can be nodulated in a NF-independent manner but what has been published suggests that this is very much dependent on soybean genotype. They mention nothing about whether their results are also very dependent on the specific soybean genotype that they used.

The data shown in Figure 2 is far from convincing. For one, they show data from nodules but clearly Nod factor signaling occurs primarily in the root epidermis during the initiation of infection and so the nodule pictures are really not relevant to the arguments of this paper. Moreover, as shown by a plethora of plant pathogen papers, there are very specific methods to convincingly show the secretion of effectors into plant cells (e.g., use of adenylyl cyclase). Cell leakage, breakage, etc could easily create artifacts that could lead to the types of pictures shown in Fig. 2.

Likewise, the whole section, starting line 239 in the manuscript, cannot be taken at face value since 1. It is all done in the heterologous tobacco system and 2. Makes use of the strong 35s promoter, which could lead to artifacts. TurboID is known to identify weak protein-protein interactions and, hence, the physiological relevance of what they report is very questionable.

Some of what they report really isn't new. For example, the fact that remorin mutations reduce nodulation has certainly been reported previously.

In short, while the authors have obviously done a great deal of work, the story is still not fully 'baked'.

Some minor comments.

1. I found it particularly irritating that the authors improperly reference their supplemental figures in the manuscript...in almost every case their numbering in the text is off 1 digit from what is actually shown in the figures. e.g., line 195, 199, 221 and many others have the supplemental figures misidentified. This suggests a general sloppy proofreading of the manuscript.

2. When comparing Figure 6a with the supplementary Figure 9a, it is clear that the relative amount of biotinylated NFR5 protein shows a similar increase in the supplementary Figure 9a, consistent with the results seen in supplementary Figure 9b. However, in the case of Figure 6a, the noticeable differences illustrated in the 6b graph are not easily discernible. To accurately capture and represent the variations depicted in the graphs, it is essential to consider replacing the current graphical representation.

3. GmREM1a plays a crucial role in recruiting GmNFR5 and GmNFR1 for the recognition of Rhizobium NF. In Figure 6, the authors suggest that NopL enhances the interaction between NFR5 and REM1. However, further experimental investigations are necessary to clarify how NopL, through REM1, enhances the NF signaling pathway. This includes determining whether it facilitates the recruitment of NFR5 to the nanodomain, where NF perception functions through REM1.

4. The authors contend that NopL plays a pivotal role in the early infection of root nodules, suggesting that this is likely facilitated by the strengthened interaction between REM1 and NFR5 through NopL. They propose that co-localization on the nodule cell membrane indicates that the enhanced interaction takes place on the nodule cell membrane. However, to clearly demonstrate its significance in early infection, it is essential to confirm whether co-localization occurs between NopL, REM1, and NFR5 in root hairs.

5. Many typos and minor mistakes:

Line 90, GmNFR5

Line 142. Inside, not insight

Line 167, Figure 3D, not 4D

Etc

6. P. 189...yes, but other remorins were identified and could be playing a role. Might explain while some nodulation is seen even in your remorin mutant.

7. I had a hard time finding the supplemental tables...they were not with the normal manuscript files. I had to download the zip file to find them. In this regard, please add the gene annotations to supplemental tables 3 and 4.

Reviewer #2 (Remarks to the Author):

The paper provides convincing data for the interaction between the T3 effector NopL and the soybean symbiotic signaling components NFR5 and SYMREM1a. Unfortunately, the paper is poorly written, and the data interpretation is often unsatisfactory.

The loss of both NopL and NodA (NF biosynthesis) does not block nodulation, clearly showing that other effectors must induce nodulation in that mutant. Knowing which effectors those are and how they act would make data interpretation more straightforward.

Nonetheless, the finding that mutation of NodQ does not enhance nodulation of the NopL mutant does suggest they act in the same pathway (activation of NF signaling). They also show a proximity labeling analysis that indicates that NopL promotes the association of NFR5 and SYMREM1a, hinting at a likely mechanism. The phenotypic evidence in Fig 1 also supports that NopL affects infection and that it makes a similar contribution to that of Nod factor signaling.

The discussion is rather unsatisfactory, as it lists many possible roles of NopL rather than critically evaluating them. Can NopL really modulate immunity, senescence, and Nod factor signaling too?

For publication in a top-ranking journal the more evidence that NopL effects NF signaling is needed (see comments on RNAseq analysis below) as well as evidence that NopL interacts with NFR5 and SYMREM1a in soybean roots.

The English grammar need substantial editing. More care needs to be taken in the writing, for example even the abbreviation of the main subject of the investigation, Nops (nodulation outer proteins) is

misrepresented as 'nodulation out proteins'. In line 79 is NtSIPK the same as SIPK in the next sentence? In line 97 subtitle, it should mention the interaction between NopL and the NF receptor rather than with the 'NF signaling cascade'. In line 316 you should say rhizobia instead of Rhizobium which is a specific genus of rhizobia.

The authors say that " the similarity between the NF and the T3E signaling pathways ". There is no T3E pathway, but T3Es can have an effect on existing signaling pathways.

The basis of the experiment is poorly explained. The authors do not mention that *Sinorhizobium fredii* HH103 is a broad host range strain, which is essential to explain why their NodA mutant can still nodulate soybean. NopL is not properly introduced, what are its properties, what do we know about its distribution across rhizobia, and its role in nodulation? Instead of introducing NopL, instead figure 1ab is devoted to NodA, which is not needed.

The authors suggest that NopL is generally important for the nodulation of soybean. This is of course, incorrect. The importance of NopL to nodulation in soybean, such as mentioned in lines 305 and 336, is limited to nodulation by *Sinorhizobium fredii* HH103. Most soybean rhizobia are NF dependent, broad host range rhizobia are exceptions.

Rather than focusing on comparing *nopL* x DN50 with *nopL* x *rem1a*, the authors should instead focus on the differences in the *nopL* mutant on DN50 vs the wt rhizobia, to see whether there is clear evidence that NF signaling is affected.

Line 81 *Phaseolus* should be italicised

Line 372. What is pairwise NF and T3 signaling?

It is not clear why authors tested whether NodA could complement the NodA Ω NopL double mutant, it would not be expected to complement under any circumstances. It does not indicate, as suggested by the authors "that NopL regulates symbiosis through the NF recognition pathway".

The authors say that the lack of additive effect of the NodA and NopL mutations suggests that they act *on* the same pathway. To me it suggests that they act *in* the same pathway (which is different than acting on the same pathway). If they act independently on the Nod factor signaling pathway, then their effect would be additive. If they act together to affect the Nod factor signaling pathway, then they would show the same phenotype. This can be tested using mutant complementation as described in the next paragraph

The abstract states: "In addition, RNA-seq analysis revealed that NopL and NF directly affect the expression of GmRINRK1, a receptor-like kinase (LRR-RLK) ortholog of the Lotus RINRK1 participating to NF signaling.

An effector could only directly affect the expression of GmRINRK1 if it acts as a transcription factor, which is not proposed here. Also, NIN also seems affected, the data show don't show a special effect on RINRK, so it is not clear why its highlighted here.

The data shows that nodulation of soybean by Sf HH103 is partly NF dependent, and partly dependent on

Line 145: Soybean nodules are determinate, so they do not have different zones like indeterminate nodules, I think you mean N-fixing cells.

The localization data is unclear, Fig 2a,b, e seems to show NopL in the cytoplasm and nucleus (not in the cell membrane as stated), but what does Fig2 d,e,f show? The control shows as much blue fluorescence as the others, and what do the arrows mean?

The authors show that overexpression of the NopL protein in rhizobia in HH103 results in an increased nodule number, but they have not tested whether this complements the NopL mutant phenotype, or the nodA mutant.

Extended figure 3. Is the body that is indicated as a 'symbiosis' a symbiosome? It does not look like one, and the picture should be zoomed out some to see the surrounding environment.

Extended data 6C, the bacterial genotypes need to be labeled on the figure

No methods are provided for the Y2H analysis, particularly Nubl should be explained

Figure 1b,c the figures are of insufficient quality

Figure 5 Should be LjNFR5?

Extended Figure 2E. The nodA mutant needs to be shown for comparison

Figure 2j. The bars below T/N/CM should be lined up with the lanes instead of being offset

Line 106 is this a soybean cultivar?

Line 161 hybrid spelled wrong

Fig 6 title. Proximity is the wrong word, you mean interaction

Fig 7 legend. The conditions (dpi) and tissue (nodule or root) for the RNAseq need to be indicated

Fig 7a. REM1a is missing

Line 270. Need to cite fig 7a here, not in the following sentence

Fig 7a. It is clear that genes in rem1a mock is lower, but other relationships are hard to interpret. Can the analysis be limited to values that are significant between certain key treatments?

Extended data figure 9. "d, Relative density of biotinylated GmNFR5 for (c)." relative to what? And what part of (c) is being referred to? Is the increase shown NopL signal relative to -NopL?

Extended data figure 9e. should be 'detection' not quantification

Line 311. It is worth mentioning that NopL is not present in the Medicago symbionts, but it also should be discussed how common this is in soybean rhizobia.

Line 332 remorininduced should be two words

Line 345-2347. It seems more likely that NopL trigger effector-triggered immunity in these genotypes

Line 353 What is GmNNL1? Without knowing what it is, it does not contribute usefully to the discussion.

Line 359. It is unclear why phosphorylation of NopL by a MAPK suggests it regulates host immunity; it seems to instead suggest its recognized/targeted by the host immune system. This needs to be explained better.

Line 706 fredii misspelled

Response to reviewer's comments:

Reviewer #1

The paper by Ma et al presents interesting data suggesting that the *R. fredii* NopL effector protein is secreted into the soybean plant and interacts with the Nod factor receptor complex to enhance nodulation. Given the role of various effector proteins in plant-pathogen interactions, the model they present is certainly plausible.

Author response: Thank you for your time and interest in our work. We have carefully considered your concerns and suggestions and have revised the manuscript accordingly. We also supplemented necessary experiments.

We believe that these changes have improved the quality of the manuscript. The following is our point-by-point response to your comments.

1. Perhaps most puzzling is how they get nodulation of soybean in a *nodA* mutant, which presumably lacks the ability to produce the Nod factor. The logic here seems faulty in that their model depends on the ability of NopL to impact Nod factor recognition and signaling but this is also true with the *nodA* mutant, where presumably there is no Nod factor. Hence, their *nodA* mutant is either quite leaky or something else is strange. I suggest they use a *nodC* mutant and see if they get similar results.

Author response 1: Thank you for your comments and useful suggestions. It is true that in many rhizobia legume interaction the production of the Nod Factor (NF) is essential for the establishment of the symbiosis. In few examples, nodulation can however be NF independent as for the *Bradyrhizobium-Aeschynomene* symbiosis (Chaintreuil et al., 2013; Okubo et al., 2012). Interestingly Okasaki (Okazaki et al., 2013) also produced a *nodC* mutant of *Bradyrhizobium* USDA61 that is partly Nod⁺ on *Glycine max* CV Enrei in agreement with our results with the *nodA* mutant. In order to confirm our result and following your suggestion, we also constructed the *S. fredii* HH103 *nodB* mutant and *nodC* mutants as well as double mutants of *NodB* or *NodC* with *NopL*, and validated the mutants using immunoblotting and qRT-PCR (Extended Data Fig. 3c-e). Nodule number of SN14 inoculated with HH103 as well as with its mutants showed that the *nodB*, *nodC* mutations were phenotypically identical to the *nodA* mutant and could still produce nodules. However, the nodule number reduced significantly compare to the WT control. This suggests that the *rhizobium*-soybean symbiosis is partly NF independent, at least in the soybean cultivars we tested. The nodulation phenotypes of the *nodA*, *nopL*, and *nodA Ω nopL* mutants are identical to the *nodB*, *nodC*, and their derived mutants, *nodB Ω nopL* and *nodC Ω nopL* (Extended Data Fig. 3a, b). In additions, there was no significant difference in the nodule number between the *nodA* mutant and the *nodB*, *nodC* mutants. These results support that NFs-deficient rhizobial mutants still produce nodules on SN14. The phenotype of the *nopL* mutant and *nopL-nodA* double mutant (or *nopL-nodB*, *nopL-nodC* double mutants) show in addition that NopL act in the NF signaling pathway and is probably required for this signaling.

The phenotype of the *nod* mutants also indicates that a NF independent signaling exist in the rhizobium-soybean interaction.

Chaintreuil, C., Arrighi, J.-F., Giraud, E., Miche, L., Moulin, L., Dreyfus, B., Munive-Hernandez, J.-A., del Carmen Villegas-Hernandez, M., and Bena, G. (2013). Evolution of symbiosis in the legume genus *Aeschynomene*. *New Phytologist* 200, 1247-1259.

Okazaki, S., Kaneko, T., Sato, S., and Saeki, K. (2013). Hijacking of leguminous nodulation signaling by the rhizobial type III secretion system. *Proceedings of the National Academy of Sciences of the United States of America* 110, 17131-17136.

Okubo, T., Fukushima, S., and Minamisawa, K. (2012). Evolution of *Bradyrhizobium-Aeschynomene* Mutualism: Living Testimony of the Ancient World or Highly Evolved State? *Plant and Cell Physiology* 53, 2000-2007.

2. Although not discussed extensively, they do mention that (as published by others), in rare cases, soybean can be nodulated in a NF-independent manner but what has been published suggests that this is very much dependent on soybean genotype. They mention nothing about whether their results are also very dependent on the specific soybean genotype that they used.

Author response 2: Thank you for your suggestions. NF-dependent nodulation may effectively depend on the soybean genotype. To check the effect of soybean genotype on the nodulation we have inoculated 15 cultivars of soybean with HH103, *nodA* mutant, *nopL* mutant, and *nodA Ω nopL* mutants. These cultivars have made significant contributions to breeding and production and have a well representative pan-genome and good genetic diversity (Liu et al., 2020). Nodule varied compared to SN14, but similarly to SN14 none of these cultivars showed a strictly NF dependent nodulation. Moreover, the nodule number of the mutants was reduced compared to WT HH103 but was not significantly different between *nodA*, *nopL* and *nodA Ω nopL* mutants (Extended Data Fig. 3f). This is again compatible with the results obtained with the *nodC* mutant in *G. max* cv. Enrei and En1282 (NF receptor mutant) plants (Okazaki et al., 2013) or *Vigna* (Nguyen et al., 2020). Therefore, we speculate that nodulation in soybean, unlike *Lotus* or *M. truncatula*, is not strictly dependent on NF (at least in these cultivars).

Nguyen, H.P., Ratu, S.T.N., Yasuda, M., Teaumroong, N., and Okazaki, S. (2020). Identification of *Bradyrhizobium elkanii* USDA61 Type III Effectors Determining Symbiosis with *Vigna mungo*. *Genes* 11.

Okazaki, S., Kaneko, T., Sato, S., and Saeki, K. (2013). Hijacking of leguminous nodulation signaling by the rhizobial type III secretion system. *Proceedings of the National Academy of Sciences of the United States of America* 110, 17131-17136.

Liu, Y., Du, H., Li, P., Shen, Y., Peng, H., Liu, S., Zhou, G.-A., Zhang, H., Liu, Z., Shi, M., et al. (2020). Pan-Genome of Wild and Cultivated Soybeans. *Cell* 182, 162-+.

3. The data shown in Figure 2 is far from convincing. For one, they show data from nodules but clearly Nod factor signaling occurs primarily in the root epidermis during the initiation of infection and so the nodule pictures are really not relevant to the arguments of this paper. Moreover, as shown by a plethora of plant pathogen papers, there are very specific methods to convincingly show the secretion of effectors into

plant cells (e.g., use of adenylyl cyclase). Cell leakage, breakage, etc could easily create artifacts that could lead to the types of pictures shown in Fig. 2.

Author response 3: Thanks to your useful suggestions. In the revised manuscript, we have supplemented the T3E-adenylyl cyclase (Cya) reporter translocation assay. Expression of NopL-Cya fusion proteins driven by the *NopL* promoter in HH103 or *ttsI* mutants. Translocation of NopL was assayed based on cAMP production by the Cya reporter in soybean roots at 1 dpi. cAMP was detected in soybean roots inoculated with HH103 (NopL-Cya) but not with wild-type HH103 or with *ttsI* (NopL-Cya) (Fig. 2b), suggesting that NopL can effectively be secreted into the root epidermis during the initiation of infection. Western Blot analysis in the root hairs (Fig. 2a), supports that the NopL can be found in the cell of the root hairs during symbiosis establishment. The Cya reporter translocation assay also proves that NopL can enter into the soybean root hairs at the early stage of infection again.

In the revised MS, we changed the order of the MS and Fig. 2 to make our presentation more logical. We first identified that NopL can be delivered into soybean root hair cells during symbiotic establishment. As NopL in NGR234 is involved in later stages of nodulation (Zhang et al., 2011), we also tested its localization in symbiotic cells of functional nodules using immunohistochemistry (IHC) and immunofluorescence (IF) analyses. These studies indicated that NopL was delivered and was predominantly present in the nitrogen-fixing cells of WT nodules (Fig. 2c-h, Extended Data Fig. 5a). These results all together show that NopL can be directly delivered into soybean root cells.

Zhang, L., Chen, X.-J., Lu, H.-B., Xie, Z.-P., and Staehelin, C. (2011). Functional Analysis of the Type 3 Effector Nodulation Outer Protein L (NopL) from *Rhizobium* sp. NGR234 SYMBIOTIC EFFECTS, PHOSPHORYLATION, AND INTERFERENCE WITH MITOGEN-ACTIVATED PROTEIN KINASE SIGNALING. *Journal of Biological Chemistry* 286, 32178-32187.

4. It is all done in the heterologous tobacco system. Makes use of the strong 35s promoter, which could lead to artifacts. TurboID is known to identify weak protein-protein interactions and, hence, the physiological relevance of what they report is very questionable.

Author response 4: Thanks to your useful suggestions. In the revised MS, we co-expressed *pGmREM1a: GmREM1a-GFP-TbID* and *GmNFR5-FLAG* with or without *NopL-Myc* in soybean hairy roots. *GFP-TbID* was co-expressed with *GmNFR5-FLAG* as a control. In this experiment, NopL promotes interactions between GmREM1a and GmNFR5 in soybean roots (Fig. 6b-d) and confirms the results obtained in the heterologous tobacco system. In addition, we also confirmed that NopL promotes the interaction between GmREM1a and GmNFR5 by semi-pull down (Fig. 6e, f). Thus, using three independent approaches we confirmed the role of NopL in the GmREM1a-GmNFR5 interaction.

5. Some of what they report really isn't new. For example, the fact that remorin mutations reduce nodulation has certainly been reported previously.

In short, while the authors have obviously done a great deal of work, the story is still not fully 'baked'.

Author response 5: This is a scientific comment. Our work aims to study the role of NopL during symbiosis and its interaction with the NF signaling. The participation of NopL in the symbiotic NF signaling is the new result. We also agree that Remorin was already described as important to symbiosis but all the work was done in *Medicago truncatula*. We thus had to validate this role in soybean as it is the first protein interacting with NopL in legume. Our work thus updates the symbiosis signaling actors in soybean and extend the function of Remorin in this system. In this revision, we have made a concerted effort to emphasize the unique aspects of our work explicitly.

6. I found it particularly irritating that the authors improperly reference their supplemental figures in the manuscript...in almost every case their numbering in the text is off 1 digit from what is actually shown in the figures. e.g., line 195, 199, 221 and many others have the supplemental figures misidentified. This suggests a general sloppy proofreading of the manuscript.

Author response 6: We are sorry for these mistakes. In the revised MS, we checked and proofread the full text for these issues and made corrections.

7. When comparing Figure 6a with the supplementary Figure 9a, it is clear that the relative amount of biotinylated NFR5 protein shows a similar increase in the supplementary Figure 9a, consistent with the results seen in supplementary Figure 9b. However, in the case of Figure 6a, the noticeable differences illustrated in the 6b graph are not easily discernible. To accurately capture and represent the variations depicted in the graphs, it is essential to consider replacing the current graphical representation.

Author response 7: Thank you for your helpful suggestions. In the revised MS, we have replaced the previous graphical representation. Compared to previous results in tobacco, the new results using soybean hairy roots to express the proteins provide better differences (Fig. 6b).

8. GmREM1a plays a crucial role in recruiting GmNFR5 and GmNFR1 for the recognition of Rhizobium NF. In Figure 6, the authors suggest that NopL enhances the interaction between NFR5 and REM1. However, further experimental investigations are necessary to clarify how NopL, through REM1, enhances the NF signaling pathway. This includes determining whether it facilitates the recruitment of NFR5 to the nanodomain, where NF perception functions through REM1.

Author response 8: Thank you for your helpful suggestions. In the revised MS, to further confirm that NopL enhances the interaction between GmREM1a and GmNFR5, and it facilitates the recruitment of GmNFR5 to the nanodomain. BiFC analysis of the interactions between GmREM1a and GmNFR5 with or without NopL showed that NopL promoted GmREM1a interaction with GmNFR5 and increased GmREM1a-YC /GmNFR5-YN clusters size in maximal projection or surface view of *N. benthamiana*

(Fig. 6g-k). This result suggests that NopL can promote GmREM1a recruit GmNFR5 to the nanodomain.

9. The authors contend that NopL plays a pivotal role in the early infection of root nodules, suggesting that this is likely facilitated by the strengthened interaction between REM1 and NFR5 through NopL. They propose that co-localization on the nodule cell membrane indicates that the enhanced interaction takes place on the nodule cell membrane. However, to clearly demonstrate its significance in early infection, it is essential to confirm whether co-localization occurs between NopL, REM1, and NFR5 in root hairs.

Author response 9: Thank you for your suggestion. To better identify the co-localization of NopL, GmNFR5, and GmREM1a, we used Western Blot to detect the NopL, GmNFR5 and GmREM1a proteins. The polyclonal antibody for GmNFR5 was produced in Rabbit. The description of this production was added in the methods section. We also identified these three proteins at the same time on the proteins isolated from the root hairs. We added this result in Fig. 6a. This experiment supports that NopL, GmREM1a, and GmNFR5 are present together in root hairs.

10. Many typos and minor mistakes:

Line 90, GmNFR5

Line 142. Inside, not insight

Line 167, Figure 3D, not 4D

Etc

Author response 10: Thank you for this. We have made corresponding modifications in the revised MS.

11. P. 189...yes, but other remorins were identified and could be playing a role. Might explain while some nodulation is seen even in your remorin mutant.

Author response 11: Thank you for your comment. We agree with you that other remorins might have a role on the nodule formation. This is also an interesting story to detect different remorins in soybean. As soybean is a paleotetraploid, there are many homologous genes of remorin in soybean. We wish we can design more detailed experiments to elucidate the function of remorin family on nodulation in future. You supply a very valuable idea for our further research study. Thank you very much!

12. I had a hard time finding the supplemental tables...they were not with the normal manuscript files. I had to download the zip file to find them. In this regard, please add the gene annotations to supplemental tables 3 and 4.

Author response 12: Sorry for this. We have added gene annotations in the supplemental tables 3, 4 and 5 of the revised MS.

Reviewer #2

The paper provides convincing data for the interaction between the T3 effector NopL and the soybean symbiotic signaling components NFR5 and SYMREM1a.

Author response: We sincerely extend our gratitude for your thoughtful comments and suggestions on our manuscript. Your concerns and suggestions have been meticulously considered, and we have modified the new manuscript accordingly.

We believe that these revisions have improved the quality of the manuscript. A list of point-by-point responses to your comments is given below.

1. Unfortunately, the paper is poorly written, and the data interpretation is often unsatisfactory.

Author response 1: We have modified the manuscript completely and the English writing was modified by a native English speaker. We hope these modifications are satisfactory for the new version.

2. The loss of both NopL and NodA (NF biosynthesis) does not block nodulation, clearly showing that other effectors must induce nodulation in that mutant. Knowing which effectors those are and how they act would make data interpretation more straightforward. Nonetheless, the finding that mutation of NodQ does not enhance nodulation of the NopL mutant does suggest they act in the same pathway (activation of NF signaling). They also show a proximity labeling analysis that indicates that NopL promotes the association of NFR5 and SYMREM1a, hinting at a likely mechanism. The phenotypic evidence in Fig 1 also supports that NopL affects infection and that it makes a similar contribution to that of Nod factor signaling.

Author response 2: We do agree that other effectors might be responsible for nodule formation in the HH103-Soybean interaction. However, there are more than 28 effectors in HH103 at least and a huge amount of work will be necessary to understand their individual role in symbiosis. In the work described here, we wished to explain the interaction between NopL and NF signaling in more detail and not to study the role of the other effectors.

3. The discussion is rather unsatisfactory, as it lists many possible roles of NopL rather than critically evaluating them. Can NopL really modulate immunity, senescence, and Nod factor signaling too?

Author response 3: In our manuscript, we showed that NopL can directly interact with NF signaling. We think that the function of NopL is multiple. From our work, we can conclude that NopL can interact with the component of NF signaling. Combining the previous work on immunity and senescence, we think it might be more complicated to study than the interaction between NopL and NFs, and might also depend on the genotype of the legume host. We added this point in the discussion and some additional references about immunity and senescence were added.

4. NopL effects on NF signaling is needed (see comments on RNAseq analysis below) as well as evidence that NopL interacts with NFR5 and SYMREM1a in soybean roots.

Author response 4: Thank you for your comments. In the revised MS, we confirmed that NopL interacts with GmREM1a and GmNFR5 by Co-IP in soybean hairy roots (Fig. 3c, Fig. 5e). We also supplemented the RNA-Seq after the *nodA* mutation to show that NopL affects NF signaling (Extended Data Fig. 12; See in author response 10).

5. The English grammar need substantial editing. More care needs to be taken in the writing, for example even the abbreviation of the main subject of the investigation, Nops (nodulation outer proteins) is misrepresented as 'nodulation out proteins'.

Author response 5: Thanks for the heads up, this was our mistake. This has been corrected, and a native English speaker also modified the English writing throughout the text.

6. In line 79 is NtSIPK the same as SIPK in the next sentence?

Author response 6: Sorry for this. NtSIPK is effectively the same as SIPK in the next sentence. We modify it accordingly.

7. In line 97 subtitle, it should mention the interaction between NopL and the NF receptor rather than with the 'NF signaling cascade'.

Author response 7: We do not completely agree with this comment because in this part of the work, the direct interaction with the NF receptor was not studied. This is the interaction with the NF signalization that is described and this is for this reason that we prefer to use NF signaling.

8. In line 316 you should say rhizobia instead of Rhizobium which is a specific genus of rhizobia.

Author response 8: Thank you for your suggestion; we have revised and updated the rhizobia throughout the revised MS.

9. The authors say that " the similarity between the NF and the T3E signaling pathways ". There is no T3E pathway, but T3Es can have an effect on existing signaling pathways.

Author response 9: Thank you for your useful suggestion. In the revised MS, we revise accordingly.

10. The basis of the experiment is poorly explained. The authors do not mention that *Sinorhizobium fredii* HH103 is a broad host range strain, which is essential to explain why their NodA mutant can still nodulate soybean.

Author response 10: Thank you for your useful suggestion. We added the introduction of *Sinorhizobium fredii* HH103 is a broad host range strain in the introduction section (Acosta-Jurado et al., 2020, Margaret et al., 2011). The broad host range behavior, like with the strain NGR234 results from the production of a complex mixture of NF. This allows the bacteria to be recognized by many hosts. However, the *nodA* mutant of this strain will be nod⁻ on every plant because lacking the core structure of the Nod factor. In this particular case broad host range will be strictly NF dependent. In the case of the *Bradyrhizobia*, effectors can also extend the host range (Camuel et

al., 2023) and this is probably what is indeed observed here in the *nodA* mutants, suggesting that effectors (but not NopL) will either hijack some later components of the NF signaling to promote symbiosis independently of the NF, or they induce symbiosis using other mechanisms/pathways. In our work we show that NopL is probably required for NF perception as its action is strictly NF dependent. This explain why the *nodA* and *nopL* mutants do not have additive effects. The revised MS was modified accordingly to this.

Acosta-Jurado, S., Alias-Villegas, C., Navarro-Gomez, P., Almozara, A., Rodriguez-Carvajal, M.A., Medina, C., and Vinardell, J.-M. (2020). *Sinorhizobium fredii* HH103 *syrM* inactivation affects the expression of a large number of genes, impairs nodulation with soybean and extends the host-range to *Lotus japonicus*. *Environmental Microbiology* 22, 1104-1124.

Camuel, A., Teulet, A., Carcagno, M., Haq, F., Pacquit, V., Gully, D., Pervent, M., Chaintreuil, C., Fardoux, J., Horta-Araujo, N., et al. (2023). Widespread *Bradyrhizobium* distribution of diverse Type III effectors that trigger legume nodulation in the absence of Nod factor. *Isme Journal* 17, 1416-1429.

Margaret, I., Becker, A., Blom, J., Bonilla, I., Goesmann, A., Goettfert, M., Lloret, J., Mittard-Runte, V., Rueckert, C., Ruiz-Sainz, J.E., et al. (2011). Symbiotic properties and first analyses of the genomic sequence of the fast growing model strain *Sinorhizobium fredii* HH103 nodulating soybean. *Journal of Biotechnology* 155, 11-19.

11. NopL is not properly introduced, what are its properties, what do we know about its distribution across rhizobia, and its role in nodulation? Instead of introducing NopL, instead figure 1ab is devoted to NodA, which is not needed.

Author response 11: This is a useful comment, we added more information on NopL in the introduction. We also supplemented the phylogenetic tree analyses of NopL in the revised MS (Extended Data Fig. 1), NopL is a conserved type III effector, which is present in a number of *Sinorhizobium* and *Bradyrhizobium* strains. However, we also believe that introducing NodA is important to explain the Nod factor signaling and effect of NopL in NF signaling.

12. The authors suggest that NopL is generally important for the nodulation of soybean. This is of course, incorrect. The importance of NopL to nodulation in soybean, such as mentioned in lines 305 and 336, is limited to nodulation by *Sinorhizobium fredii* HH103. Most soybean rhizobia are NF dependent, broad host range rhizobia are exceptions.

Author response 12: Thank you for your useful comment. In the revised MS, we used several soybean cultivars which are well representative of the soybean pan-genome and genetic diversity (Liu et al., 2020). Using these cultivars for nodulation tests support that NopL plays an important role in soybean nodulation (Extended Data Fig. 3f). Our nodulation tests show that HH103 behaves as a partially NF independent rhizobium with these soybean cultivars and NopL participates to the NF dependent

signaling through interactions with GmREM1a and GmNFR5. Therefore, NopL plays an important role in NF-dependent part of the nodulation of soybean.

Our results are in agreement with previous results obtained using *Vigna* or *G. max* (Okazaki et al., 2013; Nguyen et al., 2020). These studies showed that nodulation of these plants by the strain USDA61 is partly NF independent and also that NopL is required for efficient nodulation. The mechanisms behind these observations were not studied in these works.

Nguyen, H.P., Ratu, S.T.N., Yasuda, M., Teaumroong, N., and Okazaki, S. (2020). Identification of *Bradyrhizobium elkanii* USDA61 Type III Effectors Determining Symbiosis with *Vigna mungo*. *Genes* 11.

Okazaki, S., Kaneko, T., Sato, S., and Saeki, K. (2013). Hijacking of leguminous nodulation signaling by the rhizobial type III secretion system. *Proceedings of the National Academy of Sciences of the United States of America* 110, 17131-17136.

Liu, Y., Du, H., Li, P., Shen, Y., Peng, H., Liu, S., Zhou, G.-A., Zhang, H., Liu, Z., Shi, M., et al. (2020). Pan-Genome of Wild and Cultivated Soybeans. *Cell* 182, 162-+.

13. Rather than focusing on comparing *nopL* × DN50 with *nopL* × *rem1a*, the authors should instead focus on the differences in the *nopL* mutant on DN50 vs the wt rhizobia, to see whether there is clear evidence that NF signaling is affected.

Author response 13: Thank you for your helpful suggestions. In the revised MS, we supplemented the RNA-Seq analysis of DN50-inoculated *nodA* mutant. The *nodA* mutation produced 2658 differentially expressed genes (DEGs) compared to inoculation with wild-type HH103, and 554 DEGs were also differentially expressed in *nopL* × DN50 vs HH103 × DN50 (Extended Data Fig. 12a; Supplementary Table 4). In addition, DN50 inoculated *nodA* mutants and *nopL* mutants showed a clear positive correlation in expression levels (Pearson correlation coefficient = 0.8; $p < 2.2 \times 10^{-16}$; Extended Data Fig. 12b). Compared with HH103, inoculation with the *nopL* mutant resulted in the misexpression of a number of key symbiotic-associated genes and showed a similar expression pattern to inoculation with the *nodA* mutant (Extended Data Fig. 12c). These results provide the evidence that NopL affects NF signaling.

In addition, we also supplemented the correlation analyses between *nopL* × DN50 vs HH103 × DN50 with HH103 × *Gmrem1a* vs HH103 × DN50 and between *nopL* × *Gmrem1a* vs HH103 × DN50 with HH103 × *Gmrem1a* vs HH103 × DN50 in the revised MS. There was positive correlation in expression levels of common DEGs between *nopL* × DN50 vs HH103 × DN50 and HH103 × *Gmrem1a* vs HH103 × DN50 ($R=0.68$, Fig. 7b) and a clearer positive correlation in expression levels of common DEGs between *nopL* × *Gmrem1a* vs HH103 × DN50 and HH103 × *Gmrem1a* vs HH103 × DN50 ($R=0.98$, Fig. 7c). This also clearly suggests that NopL function is dependent on GmREM1a and this result also explains the results of the phenotype of DN50 and *Gmrem1a* inoculated *nopL* mutant.

14. Line 81 *Phaseolus* should be italicized

Author response 14: Thank you for your reminder. We have made the modifications

in the revised MS.

15. Line 372. What is pairwise NF and T3E signaling?

Author response 15: Here, we mentioned the 'pairwise' that is NF and T3E have been identified to play a pivotal to the establishment of symbiosis separately. We found that a crosstalk between NF and T3E based on the interaction between NopL and NFs. In order to have a clearer presentation of the experimental results, we have modified it accordingly in the revised MS. This original sentence has been deleted. Thank you very much!

16. It is not clear why authors tested whether NodA could complement the NodA Ω NopL double mutant, it would not be expected to complement under any circumstances. It does not indicate, as suggested by the authors "that NopL regulates symbiosis through the NF recognition pathway"

Author response 16: Thank you for the comment. The nodulation results show that nodule phenotype of $\Omega nodA$, $\Omega nopL$ and $nodA\Omega nopL$ is the same, suggesting that NopL is involved in the NF signaling. The NodA complementation in the $nodA\Omega nopL$ was to check that our single and double mutant had no (construction) problems and demonstrated that the double mutant has effectively the same symbiotic phenotype than the two single mutants showing that they act in the same pathway.

17. The authors say that the lack of additive effect of the NodA and NopL mutations suggests that they act *on* the same pathway. To me it suggests that they act *in* the same pathway (which is different than acting on the same pathway). If they act independently on the Nod factor signaling pathway, then their effect would be additive. If they act together to affect the Nod factor signaling pathway, then they would show the same phenotype. This can be tested using mutant complementation as described in the next paragraph.

Author response 17: Thank you for your very useful suggestions. We indeed modified the text in the revised MS following your suggestion. Because the nodule number is the same in the two single mutant it shows that the strength of the mutation is the same in the *nodA* and *nopL* mutants suggesting that NopL is strictly required for NF perception.

18. The abstract states: "In addition, RNA-seq analysis revealed that NopL and NF directly affect the expression of GmRINRK1, a receptor-like kinase (LRR-RLK) ortholog of the Lotus RINRK1 participating to NF signaling. An effector could only directly affect the expression of GmRINRK1 if it acts as a transcription factor, which is not proposed here. Also, NIN also seems affected, the data show don't show a special effect on RINRK, so it is not clear why its highlighted here.

Author response 18: Thank you for this comment. We modified the abstract as "In addition, RNA-seq analysis revealed that NopL and NF can influence the expression of *GmRINRK1*, a receptor-like kinase (LRR-RLK) ortholog of the *Lotus* RINRK1 participating to NF signaling. In *Lotus*, it was shown that NIN can regulate the

expression of *RINRK1* (Li et al., 2019). We here analyzed the expression of *GmNINa* and *GmRINRK1* (Fig. 7e and f). The result showed that the regulation of *GmNINa* and *GmRINRK1* are different in the wild type and *rem1a* mutant suggesting that there might exist other regulation mechanism for *GmRINRK1* in the *rem1a* mutant. It is why we highlight it here, and we added this opinion in the discussion of the revised MS.

Li, X., Zheng, Z., Kong, X., Xu, J., Qiu, L., Sun, J., Reid, D., Jin, H., Andersen, S.U., Oldroyd, G.E.D., et al. (2019). Atypical Receptor Kinase RINRK1 Required for Rhizobial Infection But Not Nodule Development in *Lotus japonicus*. *Plant Physiology* 181, 804-816.

19. The data shows that nodulation of soybean by Sf HH103 is partly NF dependent, and partly dependent on

Author response 19: Our work effectively shows that *S. fredii* HH103 nodulation is partly NF dependent. This means that NF participate to the interaction but also that symbiosis can be established, less efficiently in absence of NF.

20. Line 145: Soybean nodules are determinate, so they do not have different zones like indeterminate nodules, I think you mean N-fixing cells.

Author response 20: Thank you for your comment. We indeed mean nitrogen-fixing cells. We have modified the text in the revised version of the manuscript.

21. The localization data is unclear, Fig 2a,b, e seems to show NopL in the cytoplasm and nucleus (not in the cell membrane as stated), but what does Fig2 d,e,f show? The control shows as much blue fluorescence as the others, and what do the arrows mean?

Author response 21: The localization of NopL was determined using immunohistochemistry (IHC), but the localization of NopL in the cell membrane was not clearly visible in this experiment (Fig. 2c, d and e). In order to solve this problem, we used another more specific and complementary method, immunofluorescence (IF). In Figure 2f-h, the white arrows represent the localization of NopL (signal of Cy3, Red) in nucleus and cell membrane. Thank you for the reminder that the control did not have the same contrast as the other samples showing a stronger DAPI signal. The relevant changes have been made in the revised manuscript. Thank you again.

22. The authors show that overexpression of the NopL protein in rhizobia in HH103 results in an increased nodule number, but they have not tested whether this complements the NopL mutant phenotype, or the nodA mutant.

Author response 22: Thank you for your useful suggestions. In the revised manuscript, we did not add the phenotype analysis of overexpression of *NopL* in *nopL* mutants or *nodA* mutants. However, we added the nodule number analysis in a stably transformed transgenic soybean, expressing *NopL* from the *GmREM1a* promoter in the DN50 background (named NopL-GFP; Extended Data Fig. 8f), and inoculated with HH103, *nopL*, *nodA* and *nodA Ω NopL* mutants. The results showed that overexpression of *NopL* significantly increased the nodule number when inoculated with HH103 and *nopL*

mutant, but the nodule number did not change significantly when inoculated with *nodA* mutant or *nodA Ω nodL* mutant (Fig. 4g, h). This result indicates that in *planta* expression of *NopL* can complement the HH103 Ω *nodL* mutation and promote nodulation, except for NF producing mutants. More importantly, inoculation of HH103 and its mutants in *Gmrem1a* plants producing *NopL* does not result in a nodule number increase (Extended Data Fig. 9d, e), suggesting again that NopL affects NFs signaling through GmREM1a.

23. Extended figure 3. Is the body that is indicated as a 'symbiosis' a symbiosome? It does not look like one, and the picture should be zoomed out some to see the surrounding environment.

Author response 23: Thanks to your suggestions, In the revised manuscript, we changed the image of the symbiosome in Extended figure 5 and zoomed out appropriately to see the surrounding environment.

24. Extended data 6C, the bacterial genotypes need to be labeled on the figure

Author response 24: Thanks to your suggestions, we made the corresponding changes in the revised manuscript and added the genotypes of the bacteria.

25. No methods are provided for the Y2H analysis, particularly Nubl should be explained

Author response 25: Thanks to your suggestion, in the revised manuscript, we have supplemented the analysis and methods for Y2H in the Methods section.

26. Figure 1b,c the figures are of insufficient quality

Author response 26: Thanks to your useful suggestions. In the revised manuscript, we adjusted the quality of the images and made local zooms to highlight the ITs in Figure 1b. However, because soybean roots are thicker compared to *Lotus* or *Medicago*, we unfortunately did not have higher quality pictures of the nodule primordia under the microscope. We apologize for not modifying Figure 1c.

27. Figure 5 Should be LjNFR5?

Author response 27: Thank you for the reminder that it is not LjNFR5 but GmNFR5. To avoid ambiguity, we reworked Figure 5, and we changed the NFR5 to GmNFR5.

28. Extended Figure 2E. The *nodA* mutant needs to be shown for comparison

Author response 28: Thank you for your helpful suggestions. In the revised MS, we supplemented the nodule number phenotype of the *nodA* mutant for comparison (Extended Figure 4b).

29. Figure 2j. The bars below T/N/CM should be lined up with the lanes instead of being offset

Author response 29: Thank you for your suggestion, we have modified the problem in Figure 2a.

30. Line 106 is this a soybean cultivar?
Author response 30: Yes, Suinong 14 is a famous cultivar in Heilongjiang province which is planted in very larger area.
31. Line 161 hybrid spelled wrong
Author response 31: Thanks for the heads up, we made the changes in the revised MS.
32. Fig 6 title. Proximity is the wrong word, you mean interaction
Author response 32: Thank you for your suggestion, we made the changes in the revised MS.
33. Fig 7 legend. The conditions (dpi) and tissue (nodule or root) for the RNAseq need to be indicated
Author response 33: Thank you for your suggestion. In the revised MS, the conditions and tissue (root) has been added in the legend.
34. Fig 7a. REM1a is missing
Author response 34: This is a good question. Here, we did not detect the expression of REM1s in the transcriptome, but in the root hairs we did detect the GmREM1a protein. However, considering that only a subset of plant cells responds to and are infected by rhizobia at the whole root level and that the transcriptional response of cell types located deeper within roots, the *GmREM1a* expression cannot be easily assessed (Cervantes-Perez et al., 2022). We think that the GmREM1a is scaffold protein and it is expressed only in root hairs of soybean, not in whole roots. So, this might be the reason why we did not detect the expression of *GmREM1a* using transcriptome of the whole root. On the other hand, as observed in the phenotypic and transcriptomic results, there was more misexpression of symbiosis-associated genes in *Gmrem1a* and a reduction in the number of ITs. Consistent with our study, the results of single cell sequencing in *Medicago* (Cervantes-Perez et al., 2022) showed that SymREM1 was only expressed in a small number of cell clusters (cluster #4, 6, 9, 10 and 14 out of 25 clusters) and at very low levels after inoculation with rhizobia.
- Cervantes-Perez, S.A., Thibivilliers, S., Laffont, C., Farmer, A.D., Frugier, F., and Libault, M. (2022). Cell-specific pathways recruited for symbiotic nodulation in the *Medicago truncatula* legume. *Molecular Plant* 15, 1868-1888.
35. Line 270. Need to cite fig 7a here, not in the following sentence
Author response 35: Thanks to your suggestion, we cited Fig. 7a here.
36. Fig 7a. It is clear that genes in rem1a mock is lower, but other relationships are hard to interpret. Can the analysis be limited to values that are significant between certain key treatments?

Author response 36: Thank you for your useful suggestions. In the revised MS, we supplemented heatmap of significant differentially expressed symbiosis-related genes in DN50 inoculated with *nopL* mutant (DN50 (*ΩnopL*)) and *Gmrem1a* inoculated with (*rem1a* (HH103)) HH103 compared with DN50 inoculated with HH103 (Extended Figure 13a). There are 12 symbiosis-related DEGs in 2006 DEGs of DN50 (*ΩnopL*) and 4 symbiosis-related DEGs in *rem1a* (HH103). These symbiosis-related genes are all referenced in Roy et al. Mis-expression of these symbiosis-related genes may be responsible for the phenotypic differences.

We apologize that in the original manuscript we missed two key symbiosis-related genes *GmPLT1* (a homolog of MtPLT1) and *GmmiR172c* in the 83 DEGs. *MtPLT1* and *GmmiR172c* are key factors that coordinate the output of NF signaling required for rhizobial plant infections (Wang et al., 2019; Wang et al., 2014) or nodule development (Franssen et al., 2015). In the revised MS, we supplemented these data and results. We examined the expression of *GmPLT1* and *GmmiR172c* in WT and in the *Gmrem1a* backgrounds inoculation with either the *nodA* mutant, *nopL* mutant or *nodAΩnopL* double mutant in 24 hpi using qRT-PCR to ask whether *GmPLT1* and *GmmiR172c* affect NopL-mediated NF signaling. Although the expression of *GmPLT1* and *GmmiR172c* was lower in DN50 inoculated with HH103 compared with DN50 inoculated with its mutants, the mis-expression of *GmPLT1* and *GmmiR172c* were inconsistent with the nodule phenotype in *Gmrem1a* (Extended Figure 13b, c). These results suggest that *GmPLT1* and *GmmiR172c* is not exclusively GmREM1a dependent. This part of the supplemental results does not affect that *GmRINRK1* expression depends on a functional GmREM1a and on the NF signaling.

In addition, we retained the results of Fig. 7a, which we think it can highlight the low expression of symbiosis-related genes under mock treatment in *Gmrem1a*.

Franssen, H.J., Xiao, T.T., Kulikova, O., Wan, X., Bisseling, T., Scheres, B., and Heidstra, R. (2015). Root developmental programs shape the *Medicago truncatula* nodule meristem. *Development* 142, 2941-+.

Roy, S., Liu, W., Nandety, R.S., Crook, A., Mysore, K.S., Pislariu, C.I., Frugoli, J., Dickstein, R., and Udvardi, M.K. (2020). Celebrating 20 Years of Genetic Discoveries in Legume Nodulation and Symbiotic Nitrogen Fixation. *The Plant cell* 32, 15-41.

Wang, L., Sun, Z., Su, C., Wang, Y., Yan, Q., Chen, J., Ott, T., and Li, X. (2019). A *GmNINa-miR172c-NNC1* Regulatory Network Coordinates the Nodulation and Autoregulation of Nodulation Pathways in Soybean. *Molecular Plant* 12, 1211-1226.

Wang, Y., Wang, L., Zou, Y., Chen, L., Cai, Z., Zhang, S., Zhao, F., Tian, Y., Jiang, Q., Ferguson, B.J., et al. (2014). Soybean miR172c Targets the Repressive AP2 Transcription Factor NNC1 to Activate *ENOD40* Expression and Regulate Nodule Initiation. *Plant Cell* 26, 4782-4801.

37. Extended data figure 9. "d, Relative density of biotinylated GmNFR5 for (c)." relative to what? And what part of (c) is being referred to? Is the increase shown NopL signal

relative to -NopL?

Author response 37: This is a good question. The relative density of biotinylated GmNFR5 is the protein content of biotinylated GmNFR5 relative to the content of NopL-TbID-GFP after streptavidin Pull down. In the revised MS, we tested whether GmREM1a promotes biotinylation of GmNFR5 by NopL-GFP-TbID in soybean hairy roots. Similar to the tobacco, GmREM1a increases the ratio of the protein content of biotinylated GmNFR5 relative to the content of NopL-GFP-TbID (Relative density of biotinylated GmNFR5) in soybean.

38. Extended data figure 9e. should be 'detection' not quantification

Author response 38: Thank you for your suggestion. We have made the modifications in the manuscript.

39. Line 311. It is worth mentioning that NopL is not present in the Medicago symbionts, but it also should be discussed how common this is in soybean rhizobia.

Author response 39: Thank you for your suggestion. In the revised manuscript, we supplemented the phylogenetic tree analyses with NopL in rhizobia (Extended Data Fig. 1). NopL is specific to rhizobia (Teulet et al., 2022), and various NopL effector genes are present in a number of *Sinorhizobium* and *Bradyrhizobium* strains, including *Bradyrhizobium elkanii* USDA61, *Sinorhizobium fredii* USDA 257 and *Sinorhizobium fredii* CCBAU 25509 etc. NopL of *Sinorhizobium fredii* HH103 is the prototype of this unique T3E family. We supplemented common NopL of soybean rhizobia in results and discussion. Besides an N-terminal secretion signal sequence, NopL of HH103, which is similar to *Sinorhizobium* such as NGR234, consists of two large repeats and a C-terminal domain (Ge et al., 2016). In contrast, NopL of slow-growing Rhizobia, which lack the repeated sequence, such as USDA110 or USDA61 also interacts with GmREM1s and GmNFR5 (Response to reviewers Figure 1, not in the revised MS). The results suggest that our proposed model on NopL promoting the recruitment of GmNFR5 by GmREM1a may also be valid for slow-growing rhizobia during early symbiotic establishment in soybean.

Response to reviewers Figure 1: NopL^{USDA110} and NopL^{USDA61} interaction with GmREM1s and GmNFR5. a, b, BiFC analysis of the interactions between NopL^{USDA110} and NopL^{USDA61} with NFR5 receptors or GmREM1s in *N. benthamiana* leaves. Scale bars = 25 µm.

Ge, Y.-Y., Xiang, Q.-W., Wagner, C., Zhang, D., Xie, Z.-P., and Staehelin, C. (2016). The type 3 effector NopL of *Sinorhizobium* sp strain NGR234 is a mitogen-activated protein kinase substrate. *Journal of Experimental Botany* 67, 2483-2494.
Teulet, A., Camuel, A., Perret, X., and Giraud, E. (2022). The Versatile Roles of Type III Secretion Systems in Rhizobium-Legume Symbioses. *Annual Review of Microbiology* 76, 45-65.

40. Line 332 remorininduced should be two words

Author response 40: Thank you for your suggestion. This was our oversight and has now been corrected in the revised MS.

41. Line 345-347. It seems more likely that NopL trigger effector-triggered immunity in these genotypes

Author response 41: Thank you for your suggestion. We added discussion on this point following the comment of the reviewer. We prefer to design more experiments to detect the NopL effect on immunity in future experiments. And the selection of the soybean genotypes will be a key point for these experiments.

42. Line 353 What is GmNNL1? Without knowing what it is, it does not contribute usefully to the discussion.

Author response 42: Thank you for your suggestion. We agree and deleted the description about GmNNL1.

43. Line 359. It is unclear why phosphorylation of NopL by a MAPK suggests it regulates host immunity; it seems to instead suggest its recognized/targeted by the host immune system. This needs to be explained better.

Author response 43: Thank you for your suggestion. We modified the description accordingly.

44. Line 706 fredii misspelled

Author response 44: Thank you for your reminder. This has now been corrected in the revised manuscript.

REVIEWERS' COMMENTS

Reviewer #1 (Remarks to the Author):

The authors have added a considerable amount of additional experimental data that have addressed my major concerns. I do, however, remain concerned about the high frequency of NF independent nodulation that they report, which does not seem to be affected using different soybean genotypes. This is quite different from what has been seen in other systems, including other work in soybean. Hence, these results are quite puzzling to me. However, given the level of controls that the authors now include, I cannot criticize too strongly but remain concerned.

Some minor comments

line 80: NopA

Line 123: nodule

lines 124-125: Statement is too strong...in the vast number of cases soybean nodulation is NF dependent. Hence, my concerns about the data they are presenting.

Fig. 2f,g: really cannot see the staining they claim is present.

Reviewer #2 (Remarks to the Author):

The changes made to paper have improved the manuscript. By my assessment only one issue remains to be addressed with regards to the text.

It is widely accepted that soybean nodulation is Nod factor dependent. For instance, *B. japonicum* USDA 110 nodA mutants are completely Nod- on *Glycine max* (L.) Merr. cv. Williams (Lamb et al. below). Hence, the partly Nod factor independent nodulation reported here for Sf HH103 is an interesting and important aspect of the story, which is not properly emphasized. Also, if mentioned, readers may be given the false impression that most soybean rhizobia use NopL for nodulation. Based on this, it is critical the authors do the following:

- It should be mentioned in the abstract that the broad host range rhizobia HH103 was used

-Line 98 at the end of the abstract it should be mentioned that "We here show that NopL plays an essential role in nodulation by the broad host range rhizobia HH103"

Lamb, J.W., Hennecke, H. In *Bradyrhizobium japonicum* the common nodulation genes, nodABC, are linked to nifA and fixA . *Molec Gen Genet* 202, 512–517 (1986). <https://doi.org/10.1007/BF00333286>

Other minor changes/comments:

In line 360, I think what you mean to say is " T3E mediated secretion can affect NF signaling in soybean, with NopL playing an essential role."

role in the NF-mediated events. in which the NopL effector affects NF-mediated symbiosis signaling.

mediated symbiosis signals can act together in soybean,

In the methods, it is stated that the Donong 50 (DN50) cultivar was used, but in line 110 its says that the Suinong 14 (SN14) cultivar was used. Which was it?

Line 103 can hijack

Line 363 immune system

Line 364. Is it really in all rhizobia that nodulate soybean? If you are not sure, then you should say that " T3E NopL is broadly conserved in rhizobia that nodulate soybean.", in case there are some that do not have NopL.

Line 372-374 *S. meliloti* does not have a T3 secretion system

Response to reviewer's comments:

Reviewer #1

The authors have added a considerable amount of additional experimental data that have addressed my major concerns. I do, however, remain concerned about the high frequency of NF independent nodulation that they report, which does not seem to be affected using different soybean genotypes. This is quite different from what has been seen in other systems, including other work in soybean. Hence, these results are quite puzzling to me. However, given the level of controls that the authors now include, I cannot criticize too strongly but remain concerned.

Author response: Thank you for taking care of our manuscript and for all your comments suggesting interesting experiences for our future research work. The symbiotic interaction between soybean and rhizobium indeed depends on the genotype of soybean and rhizobium (sup Fig 3) and it will be worth in the future to study this in more details at the level of signalling. Thank you again!

Some minor comments

Line 80: NopA

Author response 1: NopA is a pilus component, which should not be considered bona fide effectors (Teulet et al., 2022). The type III effector NopAA is another Nop protein that has a catalytic domain with xyloglucanase activity and should participate in the nodulation process (Dorival et al., 2020).

Dorival, J., Phily, S., Giuntini, E., Brailly, R., de Ruyck, J., Czjzek, M., Biondi, E., and Bompard, C. (2020). Structural and enzymatic characterisation of the Type III effector NopAA (=GunA) from *Sinorhizobium fredii* USDA257 reveals a Xyloglucan hydrolase activity. *Scientific Reports* 10, 9932.

Teulet, A., Camuel, A., Perret, X., and Giraud, E. (2022). The Versatile Roles of Type III Secretion Systems in Rhizobium-Legume Symbioses. *Annual Review of Microbiology* 76, 45-65.

Line 123: nodule

Author response 2: Thank you for this. We have made the corresponding modifications in the revised MS.

Lines 124-125: Statement is too strong...in the vast number of cases soybean nodulation is NF dependent. Hence, my concerns about the data they are presenting.

Author response 3: Thank you for your comments. In the revised manuscript, we change the 'is' to 'might be', as this description is based on the experimental results. Thank you again, for your suggestion.

Fig. 2f, g: really cannot see the staining they claim is present.

Author response 4: We maximized the contrast of the NopL Cy3 signal, potentially indicating trace amounts of T3E secretion into host cells. The Cy3 signal of NopL (indicated by white arrows) is detectable in the images. Coupled with immuno-gold labelling, this

resolution supports the targetting of NopL to the cell membrane and nucleus. And we observed that overexpressed NopL exhibited a stronger signal compared to the wild type.

Reviewer #2

The changes made to paper have improved the manuscript. By my assessment only one issue remains to be addressed with regards to the text.

Author response: Thank you so much for the encouragement. We appreciate your previous and this time constructive comments. We have revised the manuscript as you suggested, and we hope it will address all your questions. Thank you again!

It is widely accepted that soybean nodulation is Nod factor dependent. For instance, *B. japonicum* USDA 110 nodA mutants are completely Nod- on *Glycine max* (L.) Merr. cv. Williams (Lamb et al. below). Hence, the partly Nod factor independent nodulation reported here for Sf HH103 is an interesting and important aspect of the story, which is not properly emphasized. Also, if mentioned, readers may be given the false impression that most soybean rhizobia use NopL for nodulation. Based on this, it is critical the authors do the following:

- It should be mentioned in the abstract that the broad host range rhizobia HH103 was used

-Line 98 at the end of the abstract it should be mentioned that "We here show that NopL plays an essential role in nodulation by the broad host range rhizobia HH103"

Lamb, J.W., Hennecke, H. In *Bradyrhizobium japonicum* the common nodulation genes, nodABC, are linked to nifA and fixA. *Molec Gen Genet* 202, 512–517 (1986). <https://doi.org/10.1007/BF00333286>

Author response 1: Thanks to your helpful suggestions. This detail is crucial to our presentation and perspective. In the revised manuscript, we have indicated in the abstract that HH103 is a broad-host rhizobia to clarify the presentation and make sure that the reader does not get confused.

Other minor changes/comments:

In line 360, I think what you mean to say is " T3E mediated secretion can affect NF signaling in soybean, with NopL playing an essential role." role in the NF-mediated events. in which the NopL effector affects NF-mediated symbiosis signaling.

mediated symbiosis signals can act together in soybean,

Author response 2: Thanks to your useful suggestions. We modified the sentence as your suggestion in the revised manuscript.

In the methods, it is stated that the Donong 50 (DN50) cultivar was used, but in line 110 it says that the Suinong 14 (SN14) cultivar was used. Which was it?

Author response 3: Thank you for your comment. In preliminary phenotypic identification, we used the cultivated soybean variety SN14. Due to the difficulty of generating transgenic soybeans in SN14, all gene editing and stable transgenic material used the DN50 variety.

Both DN50 and SN14 exhibit similar phenotypes when inoculated with HH103 or its mutants. To avoid ambiguity, we have revised the methods section accordingly.

Line 103 can hijack

Author response 4: Thank you for your suggestion. We modified the sentence as your suggestion.

Line 363 immune system

Author response 5: Thank you for this. We have made corresponding modifications in the revised MS.

Line 364. Is it really in all rhizobia that nodulate soybean? If you are not sure, then you should say that " T3E NopL is broadly conserved in rhizobia that nodulate soybean.", in case there are some that do not have NopL.

Author response 6: This is a good comment. We agree with you and included your suggestion to be more precise.

Line 372-374 S. meliloti does not have a T3 secretion system

Author response 7: This is a good comment. We modified the sentence as your suggestion. Thank you again!